# Attention Mechanisms Don't Learn Additive Models: Rethinking Feature Importance for Transformers

**Tobias Leemann** *University of Tübingen, Technical University of Munich*     *tobias.leemann@uni-tuebingen.de*

**Alina Fastowski** *Technical University of Munich*     *alina.fastowski@tum.de*

**Felix Pfeiffer** *University of Tübingen*     *felix.pfeiffer@protonmail.com*

**Gjergji Kasneci** *Technical University of Munich*     *gjergji.kasneci@tum.de*

**Reviewed on OpenReview:** *https://openreview.net/forum?id=yawWz4qWkF*

## Abstract

We address the critical challenge of applying feature attribution methods to the transformer architecture, which dominates current applications in natural language processing and beyond. Traditional attribution methods to explainable AI (XAI) explicitly or implicitly rely on linear or additive surrogate models to quantify the impact of input features on a model's output. In this work, we formally prove an alarming incompatibility: transformers are structurally incapable of representing linear or additive surrogate models used for feature attribution, undermining the grounding of these conventional explanation methodologies. To address this discrepancy, we introduce the Softmax-Linked Additive Log Odds Model (SLALOM), a novel surrogate model specifically designed to align with the transformer framework. SLALOM demonstrates the capacity to deliver a range of insightful explanations with both synthetic and real-world datasets. We highlight SLALOM's unique efficiency-quality curve by showing that SLALOM can produce explanations with substantially higher fidelity than competing surrogate models or provide explanations of comparable quality at a fraction of their computational costs. We release code for SLALOM as an open-source project online at https://github.com/tleemann/slalom_explanations.

## 1 Introduction

The transformer architecture (Vaswani et al., 2017) has been established as the status quo in modern natural language processing (Devlin et al., 2018; Radford et al., 2018; 2019; Touvron et al., 2023). However, the current and foreseeable adoption of large language models (LLMs) in critical domains such as the judicial system (Chalkidis et al., 2019) and the medical domain (Jeblick et al., 2023) comes with an increased need for transparency and interpretability. Methods to enhance the interpretability of an artificial intelligence (AI) system are developed in the research area of Explainable AI (XAI, Adadi & Berrada, 2018; Gilpin et al., 2018; Molnar, 2019; Burkart & Huber, 2021). A recent meta-study (Rong et al., 2023) shows that XAI has the potential to increase users' understanding of AI systems and their trust therein. Local feature attribution methods that quantify the contribution of each input to a decision outcome are among the most popular explanation methods, and a variety of approaches have been suggested for the task of computing such attributions (Kasneci & Gottron, 2016; Ribeiro et al., 2016; Sundararajan et al., 2017; Lundberg & Lee, 2017; Covert et al., 2021; Modarressi et al., 2022).

It remains hard to formally define the contribution of an input feature for non-linear functions. Recent work (Han et al., 2022) has shown that common explanation methods do so by implicitly or explicitly performing a local approximation of the complex black-box function, denoted as $f$, using a simpler surrogate function $g$ from a predefined class $\mathcal{G}$. For instance, Local Interpretable Model-agnostic Explanations (LIME, Ribeiro

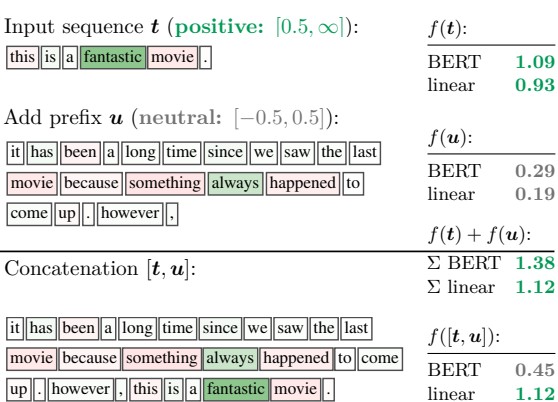

Figure 1: **Transformers cannot be well explained through additive models.** Left: We exemplarily show the log odds for the outputs of a BERT model and a linear Naïve-Bayes model ("linear") assigning each word a weight trained on the IMDB movie review dataset. The token colors indicate the weights assigned by the linear model. We pass two sequences to the models independently and in concatenation. For the linear model, the output of the concatenated sequence can be described by the sum, but this is not the case for BERT. We show that this phenomenon is not due to a non-linearity in this particular model but stems from a general incapacity of transformers to represent additive functions. Right: To overcome this difficulty, we propose SLALOM, a novel surrogate model specifically designed to better approximate transformer models.

et al., 2016) or input gradient explanations (Baehrens et al., 2010) use a linear surrogate model to approximate the black-box $f$; the model's coefficients can be used as the feature contributions.

Surrogate model explanations have the advantage that they directly describe the behavior of the model in the proximity of a specific input, i.e., under small perturbations, a property known as *fidelity* (Guidotti et al., 2018; Nauta et al., 2023). Fidelity can be quantified through the difference between the prediction model's outputs and the surrogate model's outputs (Yeh et al., 2019; Zhou et al., 2019). Explanations that quantitatively describe the prediction model's output under perturbations with low error have *high fidelity*.

An implication of models with high representative capacity and high-fidelity explanations is the *recovery property*: If the true relation $f$ between features and labels in the data is already within the function class $\mathcal{G}$, the model will learn this function and we can effectively reconstruct the original $f$ from the explanations. For example, suppose that the black-box function we consider is of linear form, i.e., $f(\boldsymbol{x}) = \boldsymbol{w}^{\top}\boldsymbol{x}$, and has been correctly learned. In this case, a gradient explanation as well as continuous LIME (C-LIME, Agarwal et al., 2021) will recover the original model's parameters up to an offset (Han et al., 2022, Theorem 1). Shapley value explanations (Lundberg & Lee, 2017) possess a comparable relationship: It is known that they correspond to the pointwise feature contributions of Generalized Additive Models (GAM, Bordt & von Luxburg, 2023). The significance of recovery properties lies in their role when explanations are leveraged to gain insights into the underlying data. Particularly when XAI is used for scientific applications such as drug discovery (Mak et al., 2023), preserving the path from the input data to the explanation through a learned model is crucial. However, such guarantees can only be provided when surrogate function class $\mathcal{G}$ can effectively mimic the model's learned relation, at least within some local region.

In this study, we demonstrate that the transformer architecture, the main building block of LLMs such as the GPT models (Radford et al., 2019), is inherently incapable of learning additive models on the input tokens, both theoretically and empirically. By additive models, we refer to models that assign each token a weight. The sum of the individual token weights then gives the output of the additive model. Linear models are a subset of this class. We formally prove that simple encoder-only and decoder-only transformers structurally cannot represent such additive models due to the attention mechanism's softmax normalization, which necessarily introduces token dependencies over the entire sequence length. An example is illustrated in Figure 1 (left). Our finding that the function spaces represented by additive models and transformers are

disjoint when dismissing trivial cases implies that prevalent feature attribution explanations *are insufficient* to model transformers. They cannot possess high fidelity, i.e., they cannot quantitatively describe these models' behavior well. This also undermines the recovery property, highlighting a significant oversight in current XAI practices. As our results suggest that the role of tokens cannot be described through a single score, we introduce the Softmax-Linked Additive Log Odds Model (SLALOM, cf. Figure 1, right), which represents the role of each input token in two dimensions: The *token value* describes the independent effect of a token, whereas the *token importance* provides a token's interaction weight when combined with other tokens in a sequence. In summary, our work offers the following contributions beyond the related literature:

(1) We theoretically and empirically demonstrate that common transformer architectures fail to represent additive and linear models on the input tokens, jeopardizing current attribution methods' fidelity.

(2) To mitigate these issues, we propose the Softmax-Linked Additive Log Odds Model (SLALOM), which uses a combination of two scores to quantify the role of input tokens.

(3) We theoretically analyze SLALOM and show that (i) it can be represented by transformers (i.e., the fidelity property), (ii) it can be uniquely identified from data (i.e., the recovery property), and (iii) it is highly efficient to estimate.

(4) Experiments on synthetic and real-world datasets with common language models (LMs) confirm the mismatch between surrogate models and predictive models, underline that two scores cover different angles of interpretability, and that SLALOM explanations can be computed that have substantially higher fidelity or efficiency than competing techniques.

## 2 Related Work

**Explainability for transformers.** Various methods exist to tackle model explainability (Molnar, 2019; Burkart & Huber, 2021). Furthermore, specific approaches have been devised for the transformer architecture (Vaswani et al., 2017): As the attention mechanism at the heart of transformer models is supposed to focus on relevant tokens, it seems a good target for explainability methods. Several works turn to attention patterns as model explanation techniques. A central attention-based method is put forward by Abnar & Zuidema (2020), who propose two methods of aggregating raw attentions across layers, *flow* and *rollout*. Brunner et al. (2020) focus on effective attentions, which aim to identify the portion of attention weights actually influencing the model's decision. While these approaches follow a scalar approach considering only attention weights, Kobayashi et al. (2020; 2023) propose a norm-based vector-valued analysis, arguing that relying solely on attention weights is insufficient and the other components of the model need to be considered. Building on the norm-based approach, Modarressi et al. (2022; 2023) further follow down the path of decomposing the transformer architecture, presenting global-level explanations with the help of rollout. Beyond that, many more attention-based explanation approaches have been put forward (Chen et al., 2020; Hao et al., 2021; Ferrando & Costa-jussà, 2021; Qiang et al., 2022; Sun et al., 2023; Yang et al., 2023) and relevance-propagation methods such as LRP have been adapted to the transformer architecture (Achtibat et al., 2024). A drawback with these model-specific explanations remains the implementational overhead that is required to adapt these methods for each architecture. On the formal side, there is no explicit method to quantitatively predict the transformer model's behavior under perturbations leaving the fidelity of these explanations unclear.

**Model-agnostic XAI.** In contrast to transformer-specific methods, researchers have devised model-agnostic explanations that can be applied without precise knowledge of a model's architecture. Model-agnostic local explanations like LIME (Ribeiro et al., 2016), SHAP (Lundberg & Lee, 2017) and others (Shrikumar et al., 2017; Sundararajan et al., 2017; Smilkov et al., 2017; Xu et al., 2020; Covert et al., 2021, etc.) are a particularly popular class of explanations that are applied to LMs as well (Szczepański et al., 2021; Schirmer et al., 2023; Dolk et al., 2022). Surrogate models are a common subform (Han et al., 2022), which locally approximate a black-box model through a simple, interpretable function.

**Linking models and explanations.** Prior work has distilled the link between classes of surrogate models that can be recovered by explanations (Agarwal et al., 2021; Han et al., 2022, Theorem 3). Notable works include Garreau & von Luxburg (2020), which provides analytical results on different parametrizations of LIME, and Bordt & von Luxburg (2023), which formalizes the connection between Shapley values and GAMs for classical Shapley values as well as $n$-Shapley values that can also model higher-order interactions.

**Interpreting feature attributions as linear models.** Many feature attribution methods do not use explicit surrogate models, e.g., SHAP or LRP. To predict model behavior under perturbations, they can be intuitively interpreted as a linear surrogate model (cf. Yeh et al., 2019): If feature $i$ has a contribution of $\phi_i$, removing $i$ should reduce the model output by $\phi_i$ (Achtibat et al., 2024), giving rise to an implicit linear model.

We contribute to the literature by theoretically showing that transformers are inherently incapable to represent additive models, casting doubts on the applicability of local LIME, SHAP, attention-based, and other attribution methods to transformers. These methods fit an explicit additive surrogate model or can be interpreted as one. Our finding that additive scores are insufficient to predict the behavior of transformers leaves us with no method that enables strict fidelity. To bridge this gap, we provide SLALOM, a novel surrogate model with substantially increased fidelity and a recovery property for transformers.

## 3 Preliminaries

### 3.1 Input and output representations

In this work, we focus on classification problems of token sequences. For the sake of simplicity, we initially consider a 2-class classification problem with labels $y \in \mathcal{Y} = \{0, 1\}$. We will outline how to generalize our approach to multi-class problems in Appendix C.1.

The input consists of a sequence of tokens $\boldsymbol{t} = [t_1, \ldots, t_{|\boldsymbol{t}|}]$ where $|\boldsymbol{t}| \in 1, \ldots, C$ is the sequence length that can span at most $C$ tokens (the context length). All tokens $t_i$ in the sequence $\boldsymbol{t}$ stem from a finite size vocabulary $\mathcal{V}$, i.e., $t_i \in \mathcal{V}, \forall i = 1, \ldots, |\boldsymbol{t}|$. To transform the tokens into a representation amenable to processing with computational methods, the tokens need to be encoded as numerical vectors. To this end, an embedding function $\boldsymbol{e} : \mathcal{V} \to \mathbb{R}^d$ is used, where $d$ is the embedding dimension. Let $\boldsymbol{e}_i = \boldsymbol{e}(t_i)$ be the embedding of the $i$-th token. The output is given by a logit vector $\boldsymbol{l} \in \mathbb{R}^{|\mathcal{Y}|}$, such that softmax($\boldsymbol{l}$) contains individual class probabilities.

Figure 2: **Transformer architecture.** In each layer $l = 1, \ldots, L$, input embeddings $\boldsymbol{h}_i^{(l-1)}$ for each token $i$ are transformed into output embeddings $\boldsymbol{h}_i^{(l)}$. When detaching the part prior to the classification head ("cls"), we see that the output only depends on the last embedding $\boldsymbol{h}_1^{(L-1)}$ and attention output $\boldsymbol{s}_1$.

### 3.2 The common transformer architecture

Many popular LMs follow the transformer architecture introduced by Vaswani et al. (2017) with only minor modifications. We will introduce the most relevant building blocks of the architecture in this section. A complete formalization is given in Appendix B.1. A schematic overview of the architecture is visualized in Figure 2. Let us denote the input embedding of token $i = 1, \ldots, |\boldsymbol{t}|$ in layer $l \in 1, \ldots, L$ by $\boldsymbol{h}_i^{(l-1)} \in \mathbb{R}^d$, where $\boldsymbol{h}_i^{(0)} = \boldsymbol{e}_i$. The core component of the attention architecture is the attention head.[1] For each token, a *query*, *key*, and a *value* vector are computed by applying an affine-linear transform to the input embeddings. Keys and queries are projected onto each other and normalized by a row-wise softmax operation resulting in attention weights $\alpha_{ij} \in [0, 1]$, denoting how much token $i$ is influenced by token $j$. The attention output for token $i$ can be computed as $\boldsymbol{s}_i = \sum_{j=1}^{|\boldsymbol{t}|} a_{ij} \boldsymbol{v}_j$, where $\boldsymbol{v}_j \in \mathbb{R}^{d_h}$ denotes the value vector for token $j$. The final $\boldsymbol{s}_i$ are projected back to dimension $d$ by a projection operator $P : \mathbb{R}^{d_h} \to \mathbb{R}^d$ before they are added to the corresponding input embedding $\boldsymbol{h}_i^{(l-1)}$ as mandated by skip-connections. The sum is then transformed by a nonlinear function that we denote by ffn : $\mathbb{R}^d \to \mathbb{R}^d$, finally resulting in a transformed embedding $\boldsymbol{h}_i^{(l)}$. This procedure is repeated iteratively for layers $1, \ldots, L$ such that we finally arrive at output embeddings $\boldsymbol{h}_i^{(L)}$. To perform classification, a classification head cls : $\mathbb{R}^d \to \mathbb{R}^{|\mathcal{Y}|}$ is put on top of a token at some index $r$ (how

---

[1]Although we only formalize a single head here, our theoretical results cover multiple heads as well

this token is chosen depends on the architecture, common choices include $r \in \{1, |\boldsymbol{t}|\}$), such that we get the final logit output $\boldsymbol{l} = \mathrm{cls}\left(\boldsymbol{h}_r^{(L)}\right)$. The logit output is transformed to a probability vector via another softmax operation. Note that in the two-class case, we obtain the log odds $F(\boldsymbol{t})$ by taking the difference ($\Delta$) between the two logits, i.e., $F(\boldsymbol{t}) \coloneqq \log \frac{p(y=1|\boldsymbol{t})}{p(y=0|\boldsymbol{t})} = \Delta(\boldsymbol{l}) = \boldsymbol{l}_1 - \boldsymbol{l}_0$.

### 3.3 Encoder-only and decoder-only models

Practical architectures can be seen as parametrizations of the process described previously. We introduce the ones relevant to this work in this section. Commonly, a distinction is made between *encoder-only* models, that include BERT (Devlin et al., 2018) and its variants, and *decoder-only* models such as the GPT models (Radford et al., 2018; 2019).

**Encoder-only models.** Considering BERT as an example of an encoder-only model, the first token is used for the classification head, i.e., $r = 1$. Usually, a special token [CLS] is prepended to the text at position 1. However this is not strictly necessary for the functioning of the model.

**Decoder-only models.** In contrast, decoder-only models like GPT-2 (Radford et al., 2019) add the classification head on top of the last token for classification, i.e., $r = |\boldsymbol{t}|$. A key difference is that in GPT-2 and other decoder-only models, a causal mask is laid over the attention matrix, resulting in $\alpha_{i,j} = 0$ for $j > i$. This encodes the constraint that tokens can only attend to themselves or to previous ones.

To make our model amenable to theoretical analysis, the transformer model in our analysis contains one slight deviation from practical models. We do not consider positional embeddings, which are added to the embeddings based on their position in the sentence. We will empirically demonstrate that using positional embeddings does not affect the validity of our findings on practical models.

## 4 Analysis

Let us initially consider a transformer with only a single layer and head. Our first insight is that the classification output can be determined only by two values: the input embedding at the classification token $r$, $\boldsymbol{h}_r^{(0)}$, and the attention output $s_r$. This can be seen when plugging in the different steps:

$$F(\boldsymbol{t}) = \Delta\left(\mathrm{cls}(\boldsymbol{h}_r^{(1)})\right) = \Delta\left(\mathrm{cls}\left(\mathrm{ffn}(\boldsymbol{h}_r^{(0)} + \boldsymbol{P}(\boldsymbol{s}_r))\right)\right) \coloneqq g(\boldsymbol{h}_r^{(0)}, \boldsymbol{s}_r) = g\left(\boldsymbol{h}_r^{(0)}, \sum_{j=1}^{|\boldsymbol{t}|} a_{rj}\boldsymbol{v}_j\right). \tag{1}$$

The attention output is given by a sum of the token value vectors $\boldsymbol{v}_j$ weighted by the respective attention weights $\alpha_{rj}$.

### 4.1 Transformers cannot represent additive models

We now consider how this architecture would represent a linear model. In this model, each token is assigned a weight $w : \mathcal{V} \to \mathbb{R}$. The output is obtained by adding weights and an offset $b \in \mathbb{R}$, consequently requiring

$$F([t_1, t_2, \ldots, t_{|\boldsymbol{t}|}]) = b + \sum_{i=1}^{|\boldsymbol{t}|} w(t_i) \tag{2}$$

for all possible input sequences.

The transformer shows surprising behavior when considering sequences of identical tokens but of different lengths, i.e., $[\tau], [\tau, \tau], \ldots$ We first note that the sum of the attention scores is bound to be $\sum_{j=1}^{|\boldsymbol{t}|} a_{rj} = 1$. The output of the attention head will thus be a weighted average of the value vectors $\boldsymbol{v}_i$. We find that the first-layer value vectors $\boldsymbol{v}_j(t_j)$ are determined purely by the input tokens $t_j$ (cf. full formalization in Appendix B.1). For a sequence of identical tokens, we will thus have the same value vectors, resulting in identical vectors being averaged. This makes the transformer produce the same output for each of these sequences. This contradicts the form in (2), where the output should successively increase by $w(t_i)$. We are now ready to state our result, which formalizes this intuition for the more general class of additive models where the token weight may also depend on its position $i$ in the input (equivalent to a GAM).

**Proposition 4.1** (Single-layer transformers cannot represent additive models.)**.** *Let $\mathcal{V}$ be a vocabulary and $C \geq 2, C \in \mathbb{N}$ be a maximum sequence length (context length). Let $w_i : \mathcal{V} \to \mathbb{R}, \forall i \in 1, ..., C$ be any map that assigns a token encountered at position $i$ a numerical score including at least one token $\tau \in \mathcal{V}$ with non-zero weight $w_i(\tau) \neq 0$ for some $i \in 2, \ldots, C$. Let $b \in \mathbb{R}$ be an arbitrary offset. Then, there exists no parametrization of the encoder or decoder single-layer transformer $F$ such that for every sequence $\boldsymbol{t} = [t_1, t_2, \ldots, t_{|\boldsymbol{t}|}]$ with length $|\boldsymbol{t}| \leq C$, the output of the transformer network is equivalent to $F([t_1, t_2, \ldots, t_{|\boldsymbol{t}|}]) = b + \sum_{i=1}^{|\boldsymbol{t}|} w_i(t_i)$.*

*Proof Sketch.* We prove the statement by concatenating the token $\tau$ to sequences of different length. We then show that the inputs to the final part $g$ of the transformer will be independent of the sequence length. Due to $g$ being deterministic, the output will also be independent of the sequence length. This is contradictory to the additive model with a weight $w_j(\tau) \neq 0$ requiring different outputs for sequences of length $j-1$ and $j$. Formal proofs for all results can be found in Appendix B. □

In simple terms, the proposition states that the transformer cannot represent any additive models on sequences of more than one token besides constant functions or those fully determined by the first input token. Importantly, the class of functions stated in the above theorem includes the prominent case of linear models in Eqn. (1), where each token has a certain weight $w$ independent of its position in the input vector (i.e., $w_i \equiv w, \forall i$, see Corollary B.2). We would like to emphasize that this statement includes the converse:

**Corollary 4.2.** *Transformers whose outputs are not constant or fully determined by the first token of the input sequence cannot be functionally equivalent to an additive model.*

### 4.2 Transformer networks with multiple layers cannot represent additive models

In this section, we will show how the argument can be extended to multi-layer transformer networks. Denote by $\boldsymbol{h}_i^{(l-1)}$ the input embedding of the $i$th token at the $l$th layer. The output is governed by the recursive relation

$$\boldsymbol{h}_i^{(l)} = \mathrm{ffn}_l(\boldsymbol{h}_i^{(l-1)} + \boldsymbol{P}_l(\boldsymbol{s}_i)) = g_l(\boldsymbol{h}_i^{(l-1)}, \boldsymbol{s}_i). \tag{3}$$

Exploiting the similar form allows us to generalize the main results to more layers recursively.

**Corollary 4.3** (Multi-Layer transformers cannot learn additive models either)**.** *Under the same conditions as in Proposition 4.1, a stack of multiple transformer blocks as the model $F$, neither has a parametrization sufficient to represent the additive model.*

**Practical considerations.** As stated earlier, the transformer model in our analysis does not consider positional embeddings that are added on the token embeddings. However, this does not have major ramifications in practice: While the transformer would be able to differentiate between sequences of different lengths with positional embeddings in theory, the softmax operation must be inverted for any input sequence by the linear feed-forward block that follows the attention mechanism. This is a highly non-linear operation and the number of possible sequences grows exponentially with the context length and vocabulary size. Learning-theoretic considerations suggest that this inversion is impossible for reasonably-sized networks as outlined in Appendix C.2. We will confirm our results with empirical findings obtained exclusively on non-modified models with positional embeddings.

## 5 A Surrogate Model for Transformers

In the previous section, we theoretically established that transformer models struggle to represent additive functions. While this must not necessarily be considered a weakness, it certainly casts doubts on the suitability of additive models as surrogate models for explanations of transformers. For a principled approach, we consider the following four requirements to be of importance:

(1) **Interpretability.** The surrogate model should be simple enough such that its parameters are inherently interpretable for humans (Molnar, 2019, Chapter 9.2).

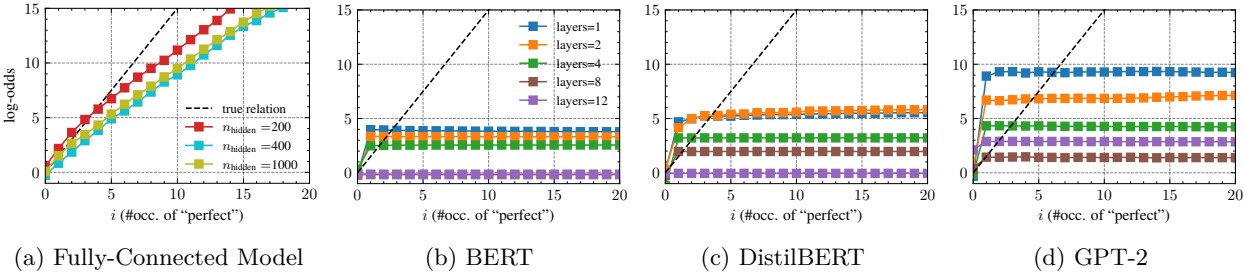

Figure 3: **Transformers fail to learn linear models.** We train different models on a synthetically sampled dataset where the log odds obey a linear relation to the features. Fully connected models (2-layer ReLU networks with different hidden layer widths) capture the linear form of the relationship well despite some estimation error (a). However, common transformer models fail to model this relationship and output almost constant values (b)-(d). This does not change with more layers.

(2) **Learnability.** The surrogate model should be easily representable by common transformers. Learnability is crucial because using a surrogate model that is hard to represent for the predictive model will likely result in low-fidelity explanations.

(3) **Recovery.** If the predictive model falls into the surrogate model's class, the fitted surrogate model's parameters should match those of the predictive model. Together with learnability, recovery ensures that a model can pick up the true relations present in the data and they can be re-identified by the explanation, which is essential, e.g., in scientific discovery with XAI.

(4) **Efficiency.** The surrogate model should be efficient to estimate even for larger models.

## 5.1 The Softmax-Linked Additive Log Odds Model

To meet the requirements, we propose a novel discriminative surrogate model that explicitly models the behavior of the softmax function. Instead of only assigning a single weight $w$ to each token, we separate two characteristics: We introduce the *token importance* as a map $s : \mathcal{V} \to \mathbb{R}$ and a *token value* in form of a map $v : \mathcal{V} \to \mathbb{R}$. Subsequently, we consider the following discriminative model:

$$F(\boldsymbol{t}) = \log \frac{p(y = 1|\boldsymbol{t})}{p(y = 0|\boldsymbol{t})} = \sum_{\tau_i \in \boldsymbol{t}} \alpha_i(\boldsymbol{t})v(t_i), \quad \text{where } \alpha_i(\boldsymbol{t}) = \frac{\exp(s(t_i))}{\sum_{t_j \in \boldsymbol{t}} \exp(s(t_j))}. \tag{4}$$

Due to the shift-invariance of the softmax function, we observe that the maps $s$ and $s'$ given by $s'(\tau) = s(\tau) + \delta$ result in the same softmax-score and thus the same log odds model for any input $\boldsymbol{t}$. Therefore, the parameterization would not be unique. To this end, we introduce a *normalization constraint* on the sum of token importances for uniqueness. Formally, we constrain it to a user-defined constant $\gamma \in \mathbb{R}$ such that $\sum_{\tau \in \mathcal{V}} s(\tau) = \gamma$, where natural choices include $\gamma \in \{0, 1\}$. We refer to the discriminative model given in Eqn. (4) together with the normalization constraint as the *softmax-linked additive log odds model* (SLALOM).

As common in surrogate model explanations, we can fit SLALOM to a predictive model's outputs globally or locally and use tuples of token importance scores and token values scores, $(v(\tau), s(\tau))$ to give explanations for an input token $\tau$. While the value score provides an absolute contribution of $\tau$ to the output, its token importance $s(\tau)$ determines its weight with respect to the other tokens. For instance, if only one token $\tau$ is present in a sequence, the output is only determined by its value score $v(\tau)$. However, in a sequence of multiple tokens, the importance of each token with respect to the others – and thereby the contribution of this token's value – is determined by the token importance scores $s$. This intuitive relation makes SLALOM interpretable, thereby satisfying Property (1).

## 5.2 Theoretical properties of SLALOM

We analyze the proposed SLALOM theoretically to ensure that it fulfills Properties (2) and (3), Learnability and Recovery, and subsequently provide efficient algorithms to estimate its parameters (4). First, we show that – unlike linear models, SLALOMs can be easily learned by transformers.

**Proposition 5.1** (Transformers can fit SLALOM). *For any maps $s$, $v$, and a transformer with an embedding size $d$ and head dimension $d_h$ with $d, d_h \geq 3$, there exists a parameterization of the transformer to reflect SLALOM in Equation* (4) *together with the normalization constraint.*

This statement can be proven by explicitly constructing the corresponding weight matrix (cf. Appendix B.5). This proposition highlights that – unlike linear models – there are simple ways for the transformer to represent relations governed by SLALOMs. We demonstrate this empirically in our experimental section and conclude that SLALOM fulfills Property (2). For Property (3), Recovery, we make the following proposition:

**Proposition 5.2** (Recovery of SLALOMs). *Suppose query access to a model $G$ that takes sequences of tokens $\boldsymbol{t}$ with lengths $|\boldsymbol{t}| \in 1, \ldots, C$ and returns the log odds according to a non-constant SLALOM on a vocabulary $\mathcal{V}$, normalization constant $\tau \in \mathbb{R}$, but with unknown parameter maps $s : \mathcal{V} \to \mathbb{R}$, $v : \mathcal{V} \to \mathbb{R}$. For $C \geq 2$, we can recover the true maps $s$, $v$ with $2|\mathcal{V}| - 1$ forward passes of $F$.*

This statement confirms property (3) and shows that SLALOM can be uniquely re-identified when we rule out the corner case of constant models. We prove it in Appendix B.6.

**Complexity considerations.** Computational complexity can be a concern for XAI methods. To estimate exact Shapley values, the model's output on exponentially many feature coalitions needs to be evaluated. However, as the proof of Proposition 5.2 shows, to estimate SLALOM's parameters for an input sequence of $\mathcal{V}$ tokens, only $2|\mathcal{V}| - 1$ forward passes are required, verifying Property (4). We empirically show that computing SLALOM explanations is about **5× faster** than computing SHAP explanations when using the same number of samples in our experimental section.

## 5.3 Numerical algorithms for computing SLALOMs

Having derived SLALOM as a better surrogate model, we require numerical algorithms to estimate $v$ and $s$. Unfortunately, the strategy derived in Proposition 5.2 using a minimal number of samples is numerically unstable. We make two key implementation choices for SLALOM to be used as an explanation technique. First, we can control the sample set of features and labels obtained through queries of the predictive model. Second, we can use different optimization strategies to fit SLALOM on this sample set. We suggest two algorithms to fit SLALOMs post-hoc on input-output pairs of a trained predictive model:

**SLALOM-`eff`**. The first version of the algorithm to fit SLALOM models is designed for maximum efficiency while maintaining reasonable performance across several XAI metrics. Obtaining a large dataset of input-output pairs can incur substantial computational costs as a forward pass of the models needs to be executed for each sample. To speed up this process, SLALOM-`Eff` uses very short sequences (we use only two tokens in this work) randomly sampled from the vocabulary for this purpose. To efficiently fit the surrogate model, we perform stochastic gradient descent on SLALOM's parameters using the mean-squared-error loss between the score output by SLALOM and the score by the predictive model as an objective. SLALOM-`eff` is our default technique used unless stated otherwise.

**SLALOM-`fidel`**. We provide another technique to fit SLALOM optimized for maximum fidelity under input perturbations such as token removals. To explain a specific sample, we sample input where we remove up to $K$ randomly sampled tokens. The sequences with tokens removed and their predictive model scores are used to fit the model, similar to LIME (Ribeiro et al., 2016). Instead of SGD, we can leverage optimizers for Least-Square-Problems to fit the parameters iteratively, however incurring a higher latency. We provide details and pseudocode for both fitting routines in Appendix D.

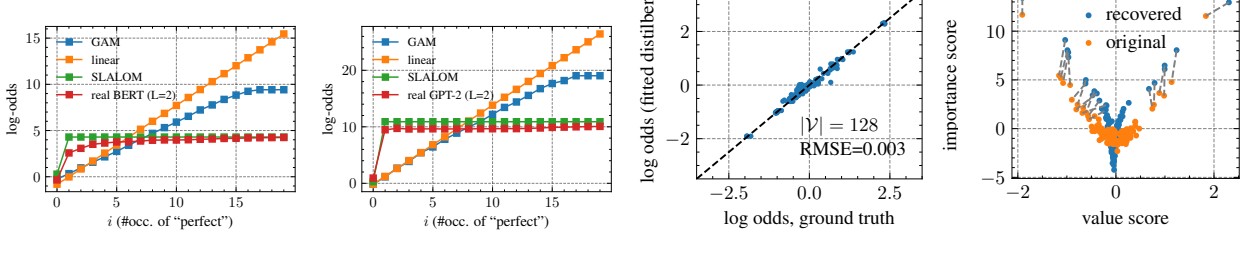

(a) Fitting SLALOM to BERT outputs

(b) Fitting SLALOM to GPT-2 outputs

(c) Comparing SLALOM and predictive output

(d) Recovering parameters of SLALOM dataset

Figure 4: **Verifying properties with synthetic data: SLALOM describes outputs of transformer models well (a, b).** We fit SLALOM to the outputs of the BERT and GPT-2 models trained on the linear synthetic dataset. The linear and GAM models (despite having $C/2{=}15\times$ more parameters) do not match the transformer's behavior. We provide another empirical counterexample and additional quantitative results in Appendix F.1. **Verifying recovery (c, d).** We verify the recovery property on a second synthetic dataset where features and labels obey a SLALOM relation. We train a 2-layer DistilBERT model on the data and fit SLALOM to the trained model. We can recover the original logit scores (c) and see a strong connection between original SLALOM parameters and the recovered ones (d). These findings verify the learnability and recovery properties. More results in Appendix F.2.

## 5.4 Relating SLALOM scores to linear attributions

Importantly, SLALOM scores can be readily converted to locally linear interpretability scores where necessary. For this purpose, a differentiable model for soft removals is required. We consider the weighted model:

$$F(\boldsymbol{\lambda}) = \frac{\sum_{t_i \in \boldsymbol{t}} \lambda_i \exp(s(t_i)) v(t_i)}{\sum_{t_i \in \boldsymbol{t}} \lambda_i \exp(s(t_i))}, \tag{5}$$

where $\lambda_i = 1$ if a token is present and $\lambda_i = 0$ if it is absent. We observe that setting $\lambda_i = 0$ has the desired effect of making the output of the soft-removal model equivalent to that of the standard SLALOM on a sequence without this token. Taking the gradients at $\boldsymbol{\lambda} = \mathbf{1}$ we obtain $\left.\frac{\partial F}{\partial \lambda_i}\right|_{\boldsymbol{\lambda}=\mathbf{1}} \propto v(t_i) \exp(s(t_i))$, which can be used to rank tokens according to the linearized attributions. We defer the derivation to Appendix B.7 and refer to these scores as *linearized* SLALOM scores. As SLALOM is just another multi-variate function, it also possible to compute SHAP values for it. We discuss this relation in Appendix B.7 as well.

## 6 Experimental Evaluation

We run a series of experiments to show the mismatch between surrogate model explanations and the transformers. Specifically, we verify that (1) real transformers fail to learn additive models, (2) SLALOM better captures transformer output, (3) SLALOM models can be recovered from fitted models with tolerable error, (4) SLALOM scores are versatile and align well with linear attribution scores and human attention, and (5) that SLALOM performs well in faithfulness metrics and has substantially higher fidelity than competing techniques. For experiments (1)-(3), we require knowledge of the ground truth and use synthetic datasets. To demonstrate the practical strengths of our method, all the experiments for (4) and (5) are conducted on real-world datasets and in comparison with state-of-the-art XAI techniques.

### 6.1 Experimental setup

**LM architectures.** We study three representative transformer language model architectures in our experiments. In sequence classification, mid-sized transformer LMs are most popular on platforms such as the Huggingface hub (Huggingface, 2023) often based on the BERT-architecture (Devlin et al., 2018), which is reflected in our experimental setup. To represent the family of encoder-only models, we deploy BERT

(Devlin et al., 2018) and DistilBERT (Sanh et al., 2019). We further experiment with GPT-2 (Radford et al., 2019), which is a decoder-only model. We use the `transformers` framework to run our experiments. While not the main scope of this work concerned with sequence classification, we will also show that due to its model-agnostic nature, SLALOM can be applied to LLMs with up to 7B parameters and non-transformer models such as Mamba (Gu & Dao, 2023) with plausible results. We provide a proof-of-concept to show that SLALOM can be applied to black-box models such as GPT-4o in Appendix F.7.

**Datasets.** We use synthetic datasets and two real-world datasets for sentiment classification. Specifically, we study the IMDB dataset, consisting of movie reviews (Maas et al., 2011) and Yelp-HAT, a dataset of restaurant reviews, for which human annotators have provided annotations on which tokens are relevant for the classification outcome (Sen et al., 2020). We provide additional details on hyperparameters, training, and datasets in Appendix E.

## 6.2 Evaluation with known ground truth

**Transformers fail to capture linear relationships.** We empirically verify the claims made in Proposition 4.1 and Corollary 4.3. To ascertain that the underlying relation captured by the models is additive, we resort to a synthetic dataset with a linear relation. The dataset is created as follows: First, we sample different sequence lengths from a binomial distribution with a mean of 15. Second, we sample words independently from a vocabulary of size 10. This vocabulary was chosen to include positive words, negative words, and neutral words, with manually assigned weights $w \in \{-1.5, -1, 0, 1, 1.5\}$, that can be used to compute a linear log odds model. We evaluate this model and finally sample the sequence label accordingly, thereby ensuring a linear relation between input sequences and log odds. We train transformer models on this dataset and evaluate them on sequences containing the same word ("perfect") multiple times. Our results in Figure 3 show that the models fail to capture the relationship regardless of the model or number of layers used. In Appendix A, we show how this undermines the recovery property with Shapley value explanations.

**Fitting SLALOM as a surrogate to transformer models.** Having demonstrated the mismatch between additive functions and transformers, we turn to SLALOM as a more suitable surrogate model. As shown in Proposition 5.1, transformers can easily fit SLALOMs, which is why we hypothesize that they should model the output of such a model well in practice. We fit the different surrogate models on a dataset of input sequences and real transformer outputs from our linear synthetic dataset and observe that linear models and additive models fail to capture the relationship learned by the transformer as shown in Figure 4(a, b). On the contrary, SALO manages to model the relationship well, even if it has considerably less trainable parameters than the additive model (GAM).

**Verifying recovery.** We run an experiment to study whether, unlike linear models, SLALOM can be fitted and recovered by transformers. To test this, we sample a second synthetic dataset that exactly follows the relation given by SLALOM. We then train transformer models on this dataset. The results in Figure 4(c, d) show that the surrogate model fitted on transformer outputs as a post-hoc explanation recovers the correct log odds mandated by SLALOM (c) and that there is a good correspondence between the true model's parameters and the recovered model's parameters (d).

## 6.3 Examining real-world predictions from different angles

We increase the difficulty and deploy SLALOM (fitted using SLALOM-`eff`) to explain predictions on real-world datasets. As there is no ground truth for these datasets, it is challenging to evaluate the quality of the explanations (Rong et al., 2022). To better understand SLALOM explanations, we study them from several angles: We compare to linear scores obtained when fitting a Naïve-Bayes Bag-of-Words (BoW) model, scores on removal and insertion benchmarks (Tomsett et al., 2020; DeYoung et al., 2020), the human attention scores available on the Yelp-HAT dataset (Sen et al., 2020), and provide qualitative results.

**Explaining Sentiment Classification.** We show qualitative results for explaining a movie review in Figure 5. The figure shows that both negative and positive words are assigned high importance scores but have value scores of different signs. Furthermore, we see that some words ("the") have positive value scores, but a very low importance. This means that they lead to positive scores on their own but are easily

| LM | values $v$ | importances $s$ | lin. |
|---|---|---|---|
| BERT | 0.619 ± 0.01 | 0.349 ± 0.01 | **0.626** ± 0.01 |
| DistilBERT | 0.692 ± 0.01 | 0.373 ± 0.01 | **0.693** ± 0.01 |
| GPT-2 | 0.618 ± 0.01 | 0.292 ± 0.01 | **0.619** ± 0.01 |
| average | 0.643 | 0.338 | **0.646** |

(a) Measuring average rank-correlation (Spearman) between Naive-Bayes scores and SLALOM scores. Linearized performs best.

| LM | values $v$ | importances $s$ | lin. |
|---|---|---|---|
| Bert | 0.786 ± 0.01 | **0.807** ± 0.01 | 0.801 ± 0.01 |
| DistilBERT | **0.688** ± 0.01 | 0.681 ± 0.01 | 0.686 ± 0.01 |
| GPT-2 | 0.674 ± 0.01 | **0.685** ± 0.01 | 0.683 ± 0.01 |
| average | 0.716 | **0.724** | 0.724 |
| BLOOM-7.1B | 0.739 ± 0.02 | 0.712 ± 0.03 | 0.740 ± 0.02 |
| Mamba-2.8B | 0.615 ± 0.03 | 0.437 ± 0.03 | 0.535 ± 0.03 |

(b) Measuring average AU-ROC between SLALOM explanations and human token attention. The importance scores and most strongly predictive of human attention for the classical models. Applying SLALOM to larger models underlines its scalability, but it remains less reliable for non-transformer models like Mamba.

| | SLALOM-`fidel` | | SLALOM-`eff` | | | | | | |
|---|---|---|---|---|---|---|---|---|---|
| LM | $v$-scores | lin. | $v$-scores | lin. | LIME | SHAP | IG | Grad | LRP |
| BERT | 0.025 ± 0.002 | **0.023** ± 0.001 | 0.031 ± 0.002 | 0.031 ± 0.002 | 0.024 ± 0.002 | 0.026 ± 0.003 | 0.557 ± 0.034 | 0.611 ± 0.033 | 0.030 ± 0.006 |
| DistilBERT | 0.028 ± 0.003 | 0.024 ± 0.002 | 0.027 ± 0.002 | 0.027 ± 0.002 | 0.027 ± 0.003 | 0.029 ± 0.003 | 0.495 ± 0.027 | 0.508 ± 0.028 | **0.023** ± 0.002 |
| GPT-2 | 0.052 ± 0.008 | 0.050 ± 0.008 | 0.089 ± 0.008 | 0.089 ± 0.008 | 0.230 ± 0.017 | **0.042** ± 0.005 | 0.454 ± 0.022 | 0.493 ± 0.023 | 0.069 ± 0.010 |
| average | 0.035 ± 0.004 | **0.032** ± 0.004 | 0.049 ± 0.004 | 0.049 ± 0.004 | 0.094 ± 0.007 | **0.032** ± 0.003 | 0.502 ± 0.028 | 0.537 ± 0.028 | 0.041 ± 0.006 |

(c) Area Over Perturbation Curve for deletion. Linearized scores and SHAP performs best in the XAI metric.

Table 1: Evaluation of SLALOM scores ("values", "importance", "lin.") with std. errors across explanation quality measures highlights that SLALOM's different scores serve different purposes. IMDB dataset shown.

overruled by other words. We compare the SLALOM scores obtained on 100 random test samples to a linear Naïve-Bayes model (obtained though counting class-wise word frequencies) as a surrogate ground truth in Table 1a through the Spearman rank correlation. We observe good agreement with the value scores $v$ and the combined linearized SLALOM scores ("lin", see Section 5.4).

**Predicting Human Attention.** To study alignment with a user perspective, we predict human attention from SLALOM scores. We compute AU-ROC for predicting annotated human attention as suggested in Sen et al. (2020) in Table 1b. We use absolute values of all signed explanations as human attention is unsigned as well. In contrast to the previous experiments, where value scores were more effective than importances, we observe that the importance scores are often best at predicting where the human attention is placed. In summary, these findings highlight that the two scores serve different purposes and cover different dimensions of interpretability. SLALOM offers higher flexibility through its 2-dimensional representation.

We also use this opportunity to study the applicability of SLALOM to larger and non-transformer models. To this end, we train Mamba and BLOOM models (Le Scao et al., 2023) with a classification head on the

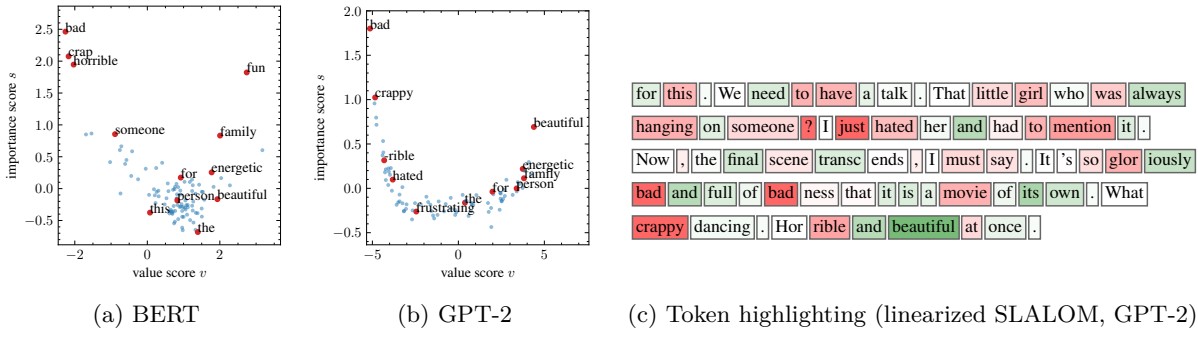

(a) BERT        (b) GPT-2        (c) Token highlighting (linearized SLALOM, GPT-2)

Figure 5: Explaining a real review with SLALOM (qualitative results). SLALOM assigns two scores to each token (a,b) and can be used to compute attributions via its linearization (c). We observe that the impactful words have high importances and the value scores indicate the sign of their contribution (positive or negative words). See Figure 12 (Appendix) for fully annotated plots.

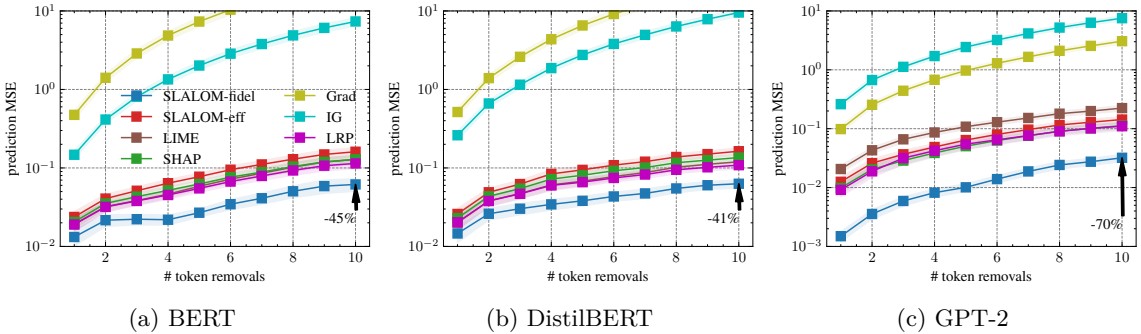

(a) BERT            (b) DistilBERT            (c) GPT-2

Figure 6: **Assessing Fidelity.** We plot the MSE for predicting model outputs under token removal and find that SLALOM's predictions have up to 70% less error than the closest competitor when up to 10 random tokens from a sentence are removed (log-$y$ plots). We interpret LRP scores as a linear model.

Yelp dataset. Our results in the lower part of Table 1b prove that SLALOM can be well applied to larger models with sizes of up to 7B. For BLOOM the results are promising and reach the same level as for the classical transformer models. SLALOM can also be applied to non-transformer models like Mamba (Gu & Dao, 2023) due to its model-agnostic nature. Due to its general expessivity, the explanations are capable of predicting human attention (at least the value scores), but we observe a slight reduction in quality. We thus see SLALOM's main scope with models that follow the classical transformer paradigm. We show that SLALOM's results for BoW correlations and human attention prediction are in the same range and often outperform competing XAI techniques in Appendix F.3.2, but defer a comparative analysis to the next sections.

**Assessing Fidelity.** Having established the roles of its components, we verify that SLALOM can produce explanations that have substantially higher fidelity than competing surrogate or non-surrogate explanation techniques. To assess this, we remove up to 10 tokens from the input sequences and use the explanations to predict the change in model output using the surrogate model (SLALOM or linear). We compare the two SLALOM versions to baselines such as LIME (Ribeiro et al., 2016), Kernel-SHAP (Lundberg & Lee, 2017), Gradients (Simonyan et al., 2013), and Integrated Gradients (IG, Sundararajan et al., 2017) and layer-wise relevance propagation (LRP) for transformers (Achtibat et al., 2024), a non-surrogate technique. We report the Mean-Squared-Error (MSE) between the predicted change and the observed change when running the model on the modified inputs in Figure 6. We observe that SLALOM-`fidel` offers substantially higher fidelity with a reduction of 70% in MSE for predicting the output changes over the second-best method (LRP). Other surrogate approaches and LRP remain cluttered together, potentially highlighting the frontier of maximum fidelity possible with a linear surrogate model.

**Evaluating XAI Metrics.** There are several other metrics to quantify the quality of explanations and to compare different explanation techniques. As a sanity check and to show that SLALOM explanations do not lag behind other techniques in established metrics, we run the classical insertion/removal benchmarks (Tomsett et al., 2020). For the insertion benchmark, we successively add the tokens with the highest attributions to the sample, which should result in a high score for the target class. We iteratively insert more tokens and compute the "Area Over the Perturbation Curve" (AOPC, see DeYoung et al. (2020)), which should be low for insertion. This metric quantifies the alignment of explanations and model behavior but only considers the feature ranking and not the assigned score. For surrogate techniques (LIME, SHAP, SLALOM) we use 5000 samples each.

| Approach | Avg. Time (s) |
|---|---|
| Grad | $0.01 \pm 0.00$ |
| IG | $0.02 \pm 0.00$ |
| LRP | $0.02 \pm 0.00$ |
| SLALOM-`eff` | $2.03 \pm 0.01$ |
| SLALOM-`fidel` | $3.77 \pm 0.24$ |
| LIME | $3.93 \pm 0.19$ |
| SHAP | $11.56 \pm 0.03$ |

Table 2: Runtime comparison using 5000 samples to estimate surrogate models.

Our results in Tab. 1c highlight that linearized SLALOM scores outperform LIME and LRP and perform on par with SHAP. Removal results for IMDB and results on YELP with similar

findings are deferred to Table 11 (Appendix). In conclusion, this shows that on top of SLALOM's desirable properties, it is on par with other techiques in common evaluation metrics.

**Computational Costs.** Finally, we take a look at the computational costs of the methods, which are mainly determined by sampling the dataset to fit the surrogate model. We provide the runtimes to explain a sample on our hardware when using 5000 samples to estimate surrogates in Table 2 with more results in Appendix F.6. We observe that SHAP incurs the highest computational burden. Among surrogate model explanations SLALOM-`eff` is the most efficient, being about $5\times$ more efficient that SHAP and $2\times$ more efficient than LIME. Nevertheless, non-surrogate techniques are far more efficient as they require only one or few (IG steps) forward or backward passes, but suffer from other disadvantages (e.g., implementation effort, no explicit way to predict model behavior). Overall, our results highlight that the two SLALOM fitting routines can produce explanations of comparable utility to other surrogate models at a fraction of the costs, or produce explanations with higher fidelity at similar costs due to structurally better alignment between surrogate and predictive models.

# 7 Discussion and Conclusion

In this work, we established that transformer networks are inherently incapable of representing linear or additive models commonly used for feature attribution. We prove that the function spaces learnable by transformers and linear models are disjoint when ruling out trivial cases. This may explain similar incapacities observed in time-series forecasting (Zeng et al., 2023), where they seem incapable of representing certain relations. To address this shortcoming, we have introduced the Softmax-Linked Additive Log Odds Model (SLALOM), a surrogate model for explaining the influence of features on transformers and other complex LMs through a two-dimensional representation.

Our work still has certain limitations that could be addressed in future work. SLALOM is specifically designed to explain the behavior of transformer models and therefore aligned with the classes of functions commonly represented by transformers. However, it would not be a suitable choice to explain models capturing a linear relationship. While SLALOM is generally applicable to any token-based LMs, we recommend using SLALOM only when the model is known to have attention-like non-linearities. Our results indicate that performance for these models is highest. We also note that SLALOM operates at the token level by assigning each individual token importance and value scores. Contextual or higher-order interpretability considering the meaning and impact of phrases, clauses, or sentences is not covered by SLALOM. To complement this theoretical foundation, future work will include further evaluation of SLALOM from a user-centric perspective, for instance, on human-centered evaluation frameworks (Colin et al., 2022). From a broader perspective, we hope that this research paves the way for advancing the interpretability and theoretical understanding of widely adopted transformer models.

**Broader Impact Statement**

This paper presents theoretical work on better understanding feature attributions in the transformer framework. We advise using caution when using our XAI technique or other model explanation as all explanations present only a simplified view of the complex ML model. Our method works best with models of the transformer architecture. Besides that, we do not see any immediate impact which we feel must be specifically highlighted here.

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

# A    Motivation: Failure Cases For Model Recovery

We provide another motivational example that shows a failure case of current explanation methods on transformer architectures. In this example we test the recovery property for a linear model. We create a synthetic dataset where each word in a sequence $t$ has a linear contribution to the log odds score, formalized by

$$\log \frac{p(y=1|\boldsymbol{t})}{p(y=0|\boldsymbol{t})} = F([t_1, t_2, \ldots, t_{|\boldsymbol{t}|}]) = b + \sum_{i=1}^{|\boldsymbol{t}|} w(t_i). \tag{6}$$

We create a dataset of 10 words (cf. Table 3) and train transformer models on samples from this dataset. We subsequently create sequences that repeatedly contain a single token $\tau$ (in this case, $\tau$="perfect"), pass them through the transformers, and use Shapley values (approximated by Kernel-Shap) to explain their output. The result is visualized in Figure 7, and shows that a fully connected model (two-layer, 400 hidden units, ReLU) recovers the correct scores, whereas transformer models fail to reflect the true relationship. This shows that explanation methods that are explicitly or implicitly based on additive models lose their ability to recover the true data-generating process when transformer models are explained.

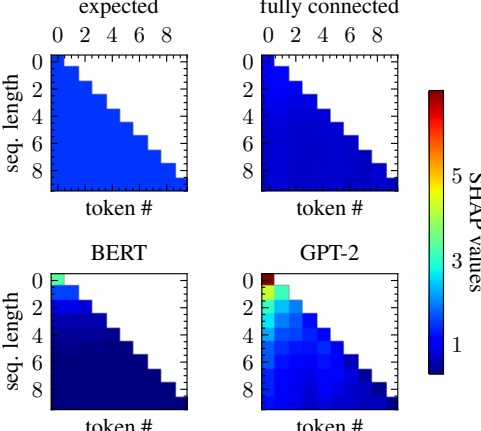

Figure 7: **SHAP values do not recover linear functions** $F$ **for transformers.** We compute SHAP values for token sequences that repeatedly contain a single token $\tau$ with a ground truth score of 1.5 (i.e., $F([\tau])$=1.5, $F([\tau, \tau])$=3.0, ...) such that the ground truth attributions should yield 1.5 independent of the sequence length. While this approximately holds true for a fully connected model, BERT and GPT-2 systematically overestimate the importance for short sequences and underestimate it for longer ones.

# B    Proofs

## B.1    Formalization of the transformer

Many popular LLMs follow the transformer architecture introduced by Vaswani et al. (2017) with only minor modifications. We will introduce the most relevant building blocks of the architecture in this section. A schematic overview of the architecture is visualized in Figure 2. Let us denote the input embeddings for Layer $l \in 1, \ldots L$ by $\boldsymbol{H}^{(l-1)} = [\boldsymbol{h}_1^{(l-1)}, \ldots, \boldsymbol{h}_{|\boldsymbol{t}|}^{(l-1)}]^\top \in \mathbb{R}^{|\boldsymbol{t}| \times d}$, where a single line $\boldsymbol{h}_i$ contains the embedding for token $i$. The input embeddings of the first layer consist of the token embeddings, i.e., $\boldsymbol{H}^{(0)} = \boldsymbol{E}$, where $\boldsymbol{E} = [\boldsymbol{e}_1, \ldots, \boldsymbol{e}_{|\boldsymbol{t}|}]^\top \in \mathbb{R}^{|\boldsymbol{t}| \times d}$ is a matrix of the individual token embeddings. At the core of the architecture lies the attention head. For each token, a *query*, *key*, and a *value* vector are computed by applying an

affine-linear transform. In matrix notation this can be written as

$$\boldsymbol{Q}^{(l)} = \boldsymbol{H}^{(l-1)}\boldsymbol{W}_Q^{(l)} + \boldsymbol{1}_{|\boldsymbol{t}|}\boldsymbol{b}_Q^{(l)^\top}, \tag{7}$$

$$\boldsymbol{K}^{(l)} = \boldsymbol{H}^{(l-1)}\boldsymbol{W}_K^{(l)} + \boldsymbol{1}_{|\boldsymbol{t}|}\boldsymbol{b}_K^{(l)^\top}, \tag{8}$$

$$\boldsymbol{V}^{(l)} = \boldsymbol{H}^{(l-1)}\boldsymbol{W}_V^{(l)} + \boldsymbol{1}_{|\boldsymbol{t}|}\boldsymbol{b}_V^{(l)^\top}, \tag{9}$$

where $\boldsymbol{1}_{|\boldsymbol{t}|} \in \mathbb{R}^{|\boldsymbol{t}|}$ denotes a vector of ones of length $|\boldsymbol{t}|$, $\boldsymbol{b}_Q^{(l)}, \boldsymbol{b}_V^{(l)}, \boldsymbol{b}_K^{(l)} \in \mathbb{R}^{d_h}$, $\boldsymbol{W}_Q^{(l)}, \boldsymbol{W}_K^{(l)}, \boldsymbol{W}_V^{(l)} \in \mathbb{R}^{d \times d_h}$, are trainable parameters and $d_h$ denotes the dimension of the attention head.[2] Keys and queries are projected onto each other and normalized by a row-wise softmax operation,

$$\boldsymbol{A}^{(l)} = \text{rowsoftmax}\left(\frac{\boldsymbol{Q}^{(l)}\boldsymbol{K}^{(l)^\top}}{\sqrt{d_k}}\right). \tag{10}$$

This results in the attention matrix $\boldsymbol{A}^{(l)} \in \mathbb{R}^{|t| \times |t|}$, where row $i$ indicates how much the other tokens will contribute to its output embedding. To compute output embeddings, we obtain attention outputs $\boldsymbol{s}_i$,

$$\boldsymbol{S} = [\boldsymbol{s}_1, \ldots, \boldsymbol{s}_{|\boldsymbol{t}|}]^\top = \boldsymbol{A}^{(l)}\boldsymbol{V}^{(l)}. \tag{11}$$

Note that an attention output can be computed as $\boldsymbol{s}_i = \sum_{j=1}^{|\boldsymbol{t}|} a_{ij}\boldsymbol{v}_j$, where $\boldsymbol{v}_j$ denotes the value vector in the line corresponding to token $j$ in $\boldsymbol{V} = [\boldsymbol{v}_1, \ldots, \boldsymbol{v}_{|\boldsymbol{t}|}]^\top$ and $a_{ij} = \boldsymbol{A}_{i,j}^{(l)}$. The final $\boldsymbol{s}_i$ are projected back to the original dimension $d$ by some projection operator $P : \mathbb{R}^{d_h} \to \mathbb{R}^d$ before they are added to the corresponding input embedding $\boldsymbol{h}_i^{(l-1)}$ due to the skip-connections. The sum is then transformed by a nonlinear function that we denote by $\text{ffn} : \mathbb{R}^d \to \mathbb{R}^d$. In summary, we obtain the output for the layer, $\boldsymbol{h}_i^{(l)}$, with

$$\boldsymbol{h}_i^{(l)} = \text{ffn}_l(\boldsymbol{h}_i^{(l-1)} + \boldsymbol{P}(\boldsymbol{s}_i)). \tag{12}$$

This procedure is repeated iteratively for layers $1, \ldots, L$ such that we finally arrive at output embeddings $\boldsymbol{H}^{(L)}$. To perform classification, a classification head $\text{cls} : \mathbb{R}^d \to \mathbb{R}^{|\mathcal{Y}|}$ is put on top of a token at classification index $r$ (how this token is chosen depends on the architecture, common choices include $r \in \{1, |\boldsymbol{t}|\}$), such that we get the final logit output $\boldsymbol{l} = \text{cls}\left(\boldsymbol{h}_r^{(L)}\right)$. The logit output is transformed to a probability vector via another softmax operation. Note that in the two-class case, we obtain the log odds $F(\boldsymbol{t})$ by taking the difference ($\Delta$) between logits

$$F(\boldsymbol{t}) \coloneqq \log\frac{p(y=1|\boldsymbol{t})}{p(y=0|\boldsymbol{t})} = \Delta(\boldsymbol{l}) = \boldsymbol{l}_1 - \boldsymbol{l}_0. \tag{13}$$

### B.2 Proof of Proposition 4.1

**Proposition B.1** (Proposition 4.1 in the main paper). *Let $\mathcal{V}$ be a vocabulary and $C \geq 2, C \in \mathbb{N}$ be a maximum sequence length (context window). Let $w_i : \mathcal{V} \to \mathbb{R}, \forall i \in 1, \ldots, C$ be any map that assigns a token encountered at position $i$ a numerical score including at least one token $\tau \in \mathcal{V}$ with non-zero weight $w_i(\tau) \neq 0$ for some $i \in 2, \ldots, C$. Let $b \in \mathbb{R}$ be an arbitrary offset. Then, there exists no parametrization of the encoder or decoder single-layer transformer $F$ such that for every sequence $\boldsymbol{t} = [t_1, t_2, \ldots, t_{|\boldsymbol{t}|}]$ with length $1 \leq |\boldsymbol{t}| \leq C$, the output of the transformer network is equivalent to*

$$F([t_1, t_2, \ldots, t_{|\boldsymbol{t}|}]) = b + \sum_{i=1}^{|\boldsymbol{t}|} w_i(t_i). \tag{14}$$

*Proof.* We show the statement in the theorem by contradiction. Consider the token, $\tau \in \mathcal{V}$, for which $w_j(\tau) \neq 0$ for some token index $j \geq 2$ which exists by the condition in the theorem. We now consider

---

[2]We only formalize one attention head here, but consider the analogous caae of multiple heads in our formal proofs.

sequences of length $k$ of the form $\boldsymbol{t}_k = [\ \underbrace{\tau, \ldots, \tau}_{\text{repeat } k \text{ times}}\ ]$ for $k = 1, \ldots, C$. For example, we have $\boldsymbol{t}_1 = [\tau]$, $\boldsymbol{t}_2 = [\tau, \tau]$, etc. The output of the transformer is given by

$$F(\boldsymbol{t}) = g\left(\boldsymbol{h}_r^{(0)}, \sum_{j=1}^{|\boldsymbol{t}|} a_{rj}\boldsymbol{v}_j\right) = g\left(\boldsymbol{e}(\tau), \sum_{j=1}^{|\boldsymbol{t}|} \alpha_{rj}\boldsymbol{v}_j\right), \tag{15}$$

where $r$ is the token index on which the classification head is placed. Note that with $r = |\boldsymbol{t}|$ for the decoder architecture, the sum always goes up to $|\boldsymbol{t}|$ (for the encoder architecture this is always true). As all tokens in the sequence have a value of $\tau$, we obtain $\boldsymbol{h}_r^{(0)} = \boldsymbol{e}(t_r) = \boldsymbol{e}(\tau)$. The first input to the final part will thus be equal for all sequences $\boldsymbol{t}_k$. We will now show that the second part will also be equal.

We compute the value, key, and query vectors for $\tau$. $\boldsymbol{v}, \boldsymbol{k}, \boldsymbol{q} \in \mathbb{R}^{d_h}$ correspond to one line in the respective key, query and value matrices. As the inputs are identical and we omit positional embeddings in this proof, all lines are identical in the matrices. This results in

$$\boldsymbol{v} = \boldsymbol{W}_V^\top \boldsymbol{e}(\tau) + \boldsymbol{b}_V \tag{16}$$

$$\boldsymbol{k} = \boldsymbol{W}_K^\top \boldsymbol{e}(\tau) + \boldsymbol{b}_K \tag{17}$$

$$\boldsymbol{q} = \boldsymbol{W}_Q^\top \boldsymbol{e}(\tau) + \boldsymbol{b}_Q \tag{18}$$

We omit the layer indices for simplicity. As pre-softmax attention scores (product of key and value vector), we obtain $s = \boldsymbol{q}^\top \boldsymbol{k}/\sqrt{d_k}$. Subsequently, the softmax computation will be performed over the entire sequence, resulting in

$$\boldsymbol{\alpha}_r = \text{softmax}([\underbrace{s, \ldots, s}_{k \text{ times}}]) = \left[\frac{\exp(s)}{k\exp(s)}\right] \tag{19}$$

$$= [\underbrace{\frac{1}{k}, \ldots, \frac{1}{k}}_{k \text{ times}}] \tag{20}$$

The second input $\sum_{j=1}^{|\boldsymbol{t}|} \alpha_{rj}\boldsymbol{v}_j$ to the feed-forward part is given by

$$\sum_{j=1}^{|\boldsymbol{t}|} \alpha_{rj}\boldsymbol{v}_j = \sum_{j=1}^{k} \alpha_{rj}\boldsymbol{v}_j = \sum_{j=1}^{k} \frac{1}{k}\boldsymbol{v} = \boldsymbol{v}, \tag{21}$$

as $\alpha_{rj}$ and $\boldsymbol{v}$ are independent of the token index $j$. We observe that the total input to final part $g$ is independent of $k$ in its entirety, as the first input $\boldsymbol{e}(\tau)$ is independent of $k$ and the second input is independent of $k$ as well. As $g$ is a deterministic function, also the log odds output will be the same for all input sequences $\boldsymbol{t}_k$ and be independent of $k$. By the condition we have a non-zero weight $w_j(\tau) \neq 0$ for some $j \geq 2$. In this case, there are two sequences $\boldsymbol{t}_{j-1}$ (length $j-1$) and $\boldsymbol{t}_j$ (length $j$) consisting of only token $\tau$, where the outputs of the additive model (GAM) follow

$$f_{\text{GAM}}(\boldsymbol{t}_j) = b + \sum_{i=1}^{j} w_i(\tau) \tag{22}$$

$$= b + \sum_{i=1}^{j-1} w_i(\tau) + w_j(\tau) \tag{23}$$

$$= f_{\text{GAM}}(\boldsymbol{t}_{j-1}) + w_j(\tau) \tag{24}$$

As we suppose $w_j(\tau) \neq 0$, it must be that $f_{\text{GAM}}(\boldsymbol{t}_j) \neq f_{\text{GAM}}(\boldsymbol{t}_{j-1})$ which is a contradiction, with the output being equal for all sequence lengths.

**Multi-head attention.** In the case of multiple heads, we have

$$F(\boldsymbol{t}) = \Delta\left(\text{cls}(\boldsymbol{h}_r^{(1)})\right) \tag{25}$$

$$= \Delta\left(\text{cls}\left(\text{ffn}(\boldsymbol{h}_r^{(0)} + \boldsymbol{P}_{h=1}(\boldsymbol{s}_r^{h=1}) + \boldsymbol{P}_{h=2}(\boldsymbol{s}_r^{h=2}) + \ldots + \boldsymbol{P}_{h=1}(\boldsymbol{s}_r^{h=H}))\right)\right) \tag{26}$$

$$= g(\boldsymbol{h}_r^{(0)}, \boldsymbol{s}_r^{h=1}, \ldots, \boldsymbol{s}_r^{h=H}) \tag{27}$$

As before, we can make the same argument, if we show that all inputs to $g$ are the same. This is straight-forward, as we can extend the argument made for one head for every head, because none of the head can differentiate between the sequence lengths. The first input will still correspond to $\boldsymbol{h}_r^{(0)} = \boldsymbol{e}(\tau)$, which results in the same contradiction. □

### B.3 Corollary: Transformers cannot represent linear models

**Corollary B.2** (Transformers cannot represent linear models). *Let the context window be $C > 2$ and suppose the same model as in Proposition 4.1. Let $w : \mathcal{V} \to \mathbb{R}$ be any weighting function that is independent of the token position with $w(\tau) \neq 0$ where for at least one token $\tau \in \mathcal{V}$. Then, the single layer transformer cannot represent the linear model*

$$F([t_1, t_2, \ldots, t_N]) = b + \sum_{i=1}^{N} w(t_i). \tag{28}$$

*Proof.* This can be seen by setting $w_i \equiv w$ for every $i$ in Proposition 4.1. With $w(\tau) \neq 0$, the condition from Proposition 4.1, i.e., having one $w_i$ with $w_i(\tau) \neq 0$ for $i \geq 2$ is fulfilled as well such that the result of the proposition as well. □

This statement has a strong implication on the capabilities of transformers as it shows that they struggle to learn linear models.

### B.4 Proof of Corollary 4.3

**Corollary B.3** (Corollary 4.3 in the main paper). *Under the same conditions as in Proposition 4.1, a stack of multiple transformer blocks as in the model $F$ neither has a parametrization sufficient to represent the additive model.*

*Proof.* We show the result by induction with the help of a lemma.

*Lemma: Suppose a set $S$ of sequences. If (1) for every sequence $\boldsymbol{t} \in S$ the input matrix $\boldsymbol{H}^{(l)} = [\boldsymbol{h}_1^{(l)}, \ldots, \boldsymbol{h}_{|\boldsymbol{t}|}^{(l)}]$ will consist of input embeddings that are identical for each token $i$, and (2) single input embeddings also have the same value for every sequence $\boldsymbol{t} \in S$, in the output $\boldsymbol{H}^{(l+1)}$ (1) the output embeddings will be identical for all tokens $i$ and (2) they will have equal value for all the sequences $\boldsymbol{t} \in S$ considered before.*

For the encoder-only architecture, the proof from Proposition 4.1 holds analogously for each token output embedding (in the previous proof, we only considered the output embedding at the classification token $r$). Without restating the full proof the main steps consist of

- considering same-token sequences of variable length

- showing the attention to be equally distributed across tokens, i.e., $\alpha_{ij} = 1/|\boldsymbol{t}|$

- showing the value vectors $\boldsymbol{v}_i$ to be equal because they only depend on the input embeddings which are equal

- concluding that the output will be equal regardless of the number of inputs

This shows that for each sequence $\boldsymbol{t} \in S$, the output at token $i$ remains constant. To show that all tokens $i$ result in the same output, we observe that the the only dependence of the input token to the output is through the query, which however is also equivalent if we have the same inputs.

For the decoder-only architecture, for token $i$, the attention weights are taken only up to index $i$ resulting in a weight of $\frac{1}{i}$ for each previous token and a weight of $0$ (masked) for subsequent ones. However, with the sum also being equal to 1 and the value vectors being equivalent, there is no difference in the outcome. This proves the lemma.

Having shown this lemma, we consider a set S of two sequences $S = \{\boldsymbol{t}_{j-1}, \boldsymbol{t}_j\}$ where $\boldsymbol{k}_{j-1}$ contains $j-1$ repetitions of token $\tau$ and $\boldsymbol{t}_j$ contains $j$ repetitions of token $\tau$. We chose $j \geq 2, \tau$ such that $w_j(\tau) \neq 0$, which is possible by the conditions of the theorem. We observe that for $\boldsymbol{H}^{(0)}$, the embeddings are equal for each token and their value is the same for both sequences. We then apply the lemma for layers $1, \ldots, L$, resulting in the output embeddings of $\boldsymbol{H}^{(L)}$ being equal for each token, and most importantly identical for $\boldsymbol{t}_{j-1}$ and $\boldsymbol{t}_j$. As we perform the classification by $F(\boldsymbol{t}) = \Delta\left(\text{cls}\left(\boldsymbol{h}_r^{(L)}\right)\right)$, this output will also not change with the sequence length. This result can be used to construct the same contradiction as in the proof of Proposition 4.1. $\square$

### B.5 Proof of Proposition 5.1

**Proposition B.4** (Transformers can easily fit SLALOM models). *For any map $s$, $v$ and a transformer with an embedding size $d, d_h \geq 3$, there exists a parameterization of the transformer to reflect the SLALOM model in Equation* (4).

*Proof.* We can prove the theorem by constructing a weight setup to reflect this map. We let the embedding $\boldsymbol{e}(\tau)$ be given by

$$\boldsymbol{e}(\tau) = [s(\tau), v(\tau), 0, 0, \ldots, 0]. \tag{29}$$

We then set the key map matrix $K$ to be

$$\boldsymbol{W}_k = \boldsymbol{0} \tag{30}$$

$$\boldsymbol{b}_k = [1, 0, \ldots, 0]. \tag{31}$$

such that we have

$$\boldsymbol{W}_k \boldsymbol{e}(\tau) + \boldsymbol{b}_k = [1, 0, \ldots, 0]. \tag{32}$$

For the query map we can use

$$\boldsymbol{W}_q = \boldsymbol{I} \tag{33}$$

$$\boldsymbol{b}_q = \boldsymbol{0} \tag{34}$$

such that

$$\boldsymbol{W}_v \boldsymbol{e}(\tau) + \boldsymbol{b}_v = [s(\tau), v(\tau), 0, \ldots, 0]. \tag{35}$$

This results in the non-normalized attention scores for query $\tau \in \mathcal{V}$ and key $\theta \in \mathcal{V}$

$$a(t_i, t_j) = (\boldsymbol{W}_q \boldsymbol{e}(t_i) + \boldsymbol{b}_q)^\top (\boldsymbol{W}_k \boldsymbol{e}(t_j) + \boldsymbol{b}_k) = s(t_j) \tag{36}$$

We see that regardless of the query token, the pre-softmax score will be $s(\theta)$. For the value scores, we perform a similar transform with

$$\boldsymbol{W}_v = \text{diag}\left([0, 1, 0, \ldots, 0]\right) \tag{37}$$

$$\boldsymbol{b}_v = \boldsymbol{0} \tag{38}$$

such that

$$\boldsymbol{v}_i = \boldsymbol{W}_v \boldsymbol{e}(t_i) + \boldsymbol{b}_v = [0, 0, v(t_i), 0, \ldots, 0]. \tag{39}$$

We then obtain

$$\boldsymbol{s}_r = \sum_{t_i \in \boldsymbol{t}} a_{ri} \boldsymbol{v}_i = \sum_{t_i \in \boldsymbol{t}} \mathrm{softmax}_i [s(t_1), \ldots, s(t_{|\boldsymbol{t}|})] \boldsymbol{v}_i \tag{40}$$

$$= \left[ 0, 0, \sum_{t_i \in \boldsymbol{t}} \alpha_i(\boldsymbol{t}) v(t_i), \ldots, 0 \right]^\top \tag{41}$$

We saw that the final output can be represented by

$$F(\boldsymbol{t}) = \Delta \left( \mathrm{cls} \left( \mathrm{ffn} \left( \boldsymbol{e}(t_0) + P(\boldsymbol{s}_r) \right) \right) \right) \tag{42}$$

The projection operator is linear, which can set to easily forward in input by setting $\boldsymbol{P} \equiv \boldsymbol{I}$. Due to the skip connection of the feed-forward part, we can easily transfer the second part through the first ffn part. In the classification part, we output the third component and zero by applying the final weight matrix

$$\boldsymbol{W}_{class} = \begin{bmatrix} 0 & 0 & 0 & \cdots & 0 \\ 0 & 0 & 1 & \cdots & 0 \end{bmatrix} \tag{43}$$

and a bias vector of $\boldsymbol{0}$.

**Multiple Heads.** Multiple heads can represent the pattern by choosing $P = \boldsymbol{I}$ for one head and choosing $P = \boldsymbol{0}$ for the other heads.

**Multiple Layers.** We can extend the argument to multiple layers by showing that the input vectors can just be forwarded by the transformer. This is simple by setting $\boldsymbol{P} \equiv \boldsymbol{0}$, the null-map, which can be represented by a linear operator. We then use the same classification hat as before. $\qquad\square$

### B.6 Proof of Proposition 5.2

**Proposition B.5** (Proposition 5.2. in the main paper). *Suppose query access to a model $G$ that takes sequences of tokens $\boldsymbol{t}$ with lengths $|\boldsymbol{t}| \in 1, \ldots, C$ and returns the log odds according to a non-constant SLALOM on a vocabulary $\mathcal{V}$ with unknown parameter maps $s : \mathcal{V} \to \mathbb{R}$, $v : \mathcal{V} \to \mathbb{R}$. For $C \geq 2$, we can recover the true maps $s$, $v$ with $2|\mathcal{V}| - 1$ queries (forward passes) of $F$.*

*Proof.* We first compute $G([\tau]), \forall \tau \in \mathcal{V}$. We know that for single token sequences, all attention is on one token, i.e., $(\alpha_i = 1)$ and we thus have

$$G([\tau]) = v(\tau) \tag{44}$$

We have obtained the values scores $v$ for each token through $|\mathcal{V}|$ forward passes. To identify the token importance scores $s$, we consider token sequences of length 2.

We first note that if the SLALOM is non-constant and $|\mathcal{V}| > 1$, for every token $\tau \in \mathcal{V}$, we can find another token $\theta$ for which $v(\tau) \neq v(\theta)$. This can be seen by contradiction: If this would not be the case, i.e., we cannot find a token $\omega$ with a different value $v(\omega)$, all tokens have the same value and the SLALOM would have to be constant. For $|\mathcal{V}| = 1$, SLALOM is always constant and does not fall under the conditions of the theorem.

We now select an arbitrary reference token $\theta \in \mathcal{V}$. We select another token $\hat{\theta}$ for which $v(\hat{\theta}) \neq v(\theta)$. By the previous argument such a token always exists if the SLALOM is non-constant. We now compute relative importances w.r.t. $\theta$ that we refer to as $\eta_\theta$. We let $\eta_\theta(\tau) = s(\tau) - s(\theta)$ denote the difference of the importance between the importance scores of tokens $\tau, \theta \in \mathcal{V}$. We set $\eta_\theta(\theta) = 0$

We start with selecting token $\tau = \hat{\theta}$ and subsequently use each other token $\tau \neq \theta$ to perform the following steps **for each** $\tau \neq \theta$:

**1. Identify reference token $\hat{\tau}$.** We now have to differentiate two cases: If $v(\tau) = v(\theta)$, we select $\hat{\tau} = \hat{\theta}$ as the reference token. If $v(\tau) \neq v(\theta)$, we select $\hat{\tau} = \theta$ as the reference token. By doing so, we will always have $v(\hat{\tau}) \neq v(\tau)$.

**2. Compute $G([\tau, \hat{\tau}])$.** We now compute $G([\tau, \hat{\tau}])$. From the model's definition, we know that

$$G([\tau, \hat{\tau}]) = \frac{\exp(s(\tau))}{\exp(s(\tau)) + \exp(s(\hat{\tau}))} v(\tau) \tag{45}$$

$$+ \frac{\exp(s(\hat{\tau}))}{\exp(s(\tau)) + \exp(s(\hat{\tau}))} v(\hat{\tau}) \tag{46}$$

$$= \frac{\exp(s(\tau))}{\exp(s(\tau)) + \exp(s(\hat{\tau}))} G([\tau]) \tag{47}$$

$$+ \frac{\exp(s(\hat{\tau}))}{\exp(s(\tau)) + \exp(s(\theta))} G([\hat{\tau}]) \tag{48}$$

$$= \frac{\exp(s(\tau))}{\exp(s(\tau)) + \exp(s(\hat{\tau}))} G([\tau])) \tag{49}$$

$$+ \left(1 - \frac{\exp(s(\tau))}{\exp(s(\tau)) + \exp(s(\hat{\tau}))}\right) G([\hat{\tau}]) \tag{50}$$

which we can rearrange to

$$G([\tau, \hat{\tau}]) - G([\hat{\tau}]) = \frac{\exp(s(\tau))}{\exp(s(\tau)) + \exp(s(\hat{\tau}))} \left(G([\tau]) - G([\hat{\tau}])\right) \tag{51}$$

and finally to

$$\frac{\exp(s(\tau))}{\exp(s(\tau)) + \exp(s(\hat{\tau}))} = \frac{G([\tau, \hat{\tau}]) - G([\hat{\tau}])}{G([\tau]) - G([\hat{\tau}])} := g(\tau, \hat{\tau}) \tag{52}$$

and

$$\frac{1}{1 + \frac{\exp(s(\hat{\tau}))}{\exp(s(\tau))}} = g(\tau, \hat{\tau}) \tag{53}$$

$$\Leftrightarrow \frac{1}{g(\tau, \hat{\tau})} = 1 + \frac{\exp(s(\hat{\tau}))}{\exp(s(\tau))} \tag{54}$$

$$\Leftrightarrow \log\left(\frac{1}{g(\tau, \hat{\tau})} - 1\right) = s(\tau) - s(\hat{\tau}) := d(\tau, \hat{\tau}) \tag{55}$$

This allows us to express the importance of every token $\tau \in \mathcal{V}$ relative to the base token $\hat{\tau}$.

**3. Compute importance relative to $\theta$, i.e., $\eta_\theta(\tau)$.** In case we selected $\hat{\tau} = \theta$, we set $\eta_\theta(\tau) = d(\tau, \hat{\tau}) = s(\tau) - s(\theta)$. In case we selected $\hat{\tau} = \hat{\theta}$, we set

$$\eta_\theta(\tau) = d(\tau, \hat{\tau}) - d(\hat{\theta}, \theta) = s(\tau) - s(\hat{\theta}) + (s(\hat{\theta}) - s(\theta)) = s(\tau) - s(\theta) \tag{56}$$

The value of $d(\hat{\theta}, \theta)$ is already known from the first iteration of the loop, where we consider $\tau = \hat{\theta}$ (and needs to be computed only once).

Having obtained a value of $\eta_\theta(\tau)$ for each token $\tau \neq \theta$, with $|\mathcal{V}| - 1$ forward passes, we can then use the normalization in constraint to solve for $s(\theta)$ as in

$$\sum_{\tau \in \mathcal{V}} (\eta_\theta(\tau) + s(\theta)) \stackrel{!}{=} 0 \tag{57}$$

such that we obtain

$$s(\theta) = \frac{\sum_{\tau \in \mathcal{V}} \eta_\theta(\tau)}{|\mathcal{V}|} \tag{58}$$

We can plug this back in to obtain the values for all token importance scores $s(\tau) = s(\theta) + \eta_\theta(\tau)$. We have thus computed the maps $s$ and $v$ in $2|\mathcal{V}| - 1$ forward passes, which completes the proof. □

### B.7 Relating SLALOM to other attribution techniques.

**Local Linear Attribution Scores.** We can consider the following weighted model:

$$F(\boldsymbol{\lambda}) = \frac{\sum_{t_i \in \boldsymbol{t}} \lambda_i \exp(s(t_i)) v(t_i)}{\sum_{t_i \in \boldsymbol{t}} \lambda_i \exp(s(t_i))} \tag{59}$$

where $\lambda_i = 1$ if a token is present and $\lambda_i = 0$ if it is absent. We observe that setting $\lambda_i = 0$ has the desired effect of making the output of the weighted model equivalent to that of the unweighted SLALOM on a sequence without this token.

Taking the derivative at $\boldsymbol{\lambda} = \mathbf{1}$ results in

$$\frac{\partial F}{\partial \lambda_i} = \frac{\exp(s(t_i)) v(t_i) \left( \sum_{t_j \in \boldsymbol{t}, j \neq i} \lambda_j \exp(s(t_j)) \right)}{\left( \sum_{t_j \in \boldsymbol{t}} \lambda_j \exp(s(t_j)) \right)^2} \tag{60}$$

$$- \frac{\exp(s(t_i)) \left( \sum_{t_j \in \boldsymbol{t}, j \neq i} \lambda_j \exp(s(t_j)) v(t_j) \right)}{\left( \sum_{t_j \in \boldsymbol{t}} \lambda_j \exp(s(t_j)) \right)^2} \tag{61}$$

Plugging in $\boldsymbol{\lambda} = \mathbf{1}$, and using $\alpha_i(\boldsymbol{t}) = \frac{\exp(s(t_i))}{\sum_{t_j \in \boldsymbol{t}} \lambda_j \exp(s(t_j))}$ we obtain

$$\left. \frac{\partial F}{\partial \lambda_i} \right|_{\boldsymbol{\lambda}=\mathbf{1}} = \alpha_i \left( v(t_i)(1 - \alpha_i(\boldsymbol{t})) - (F(\mathbf{1}) - \alpha_i v(t_i)) \right) \tag{62}$$

$$= \alpha_i \left( (v(t_i) - \alpha_i v(t_i))) - (F(\mathbf{1}) - \alpha_i v(t_i)) \right) \tag{63}$$

$$= \alpha_i \left( v(t_i) - F(\mathbf{1}) \right) \tag{64}$$

Noting that $\alpha_i = \frac{\exp(s(t_i))}{R}$, where $R$ and $F(\mathbf{1})$ are independent of $i$, we obtain

$$\left. \frac{\partial F}{\partial \lambda_i} \right|_{\boldsymbol{\lambda}=\mathbf{1}} \propto v(t_i) \exp(s(t_i)), \tag{65}$$

which can be used to rank tokens according the locally linear attributions. We refer to this expression as linearized SLALOM scores ("lin").

**Shapley Values.** We can convert SLALOM scores to Shapley values $\phi(i)$ using the explicit formula:

$$\phi(i) = \frac{1}{n} \sum_{S \subset [N] \setminus \{i\}} \binom{n-1}{|S|} (F(S \cup \{i\}) - F(S)) \tag{66}$$

$$= \frac{1}{n} \sum_{S \subset [N] \setminus \{i\}} \binom{n-1}{|S|} \left( F(S \cup \{i\}) - \frac{F(S \cup \{i\}) - \alpha_i v_i}{1 - \alpha_i} \right) \tag{67}$$

$$= \frac{1}{n} \sum_{S \subset [N] \setminus \{i\}} \binom{n-1}{|S|} \left( \frac{\alpha_i(v_i - F(S \cup \{i\}))}{1 - \alpha_i} \right) \tag{68}$$

$$\left( \frac{\alpha_i(v_i - F(\mathbf{1}))}{1 - \alpha_i} \right) \tag{69}$$

However, computing this sum remains usually intractable, as the number of coalitions grows exponentially. We can resort to common sampling approaches (Castro et al., 2009; Maleki et al., 2013) to approximate the sum.

## C  Additional Discussion and Intuition

### C.1  Generalization to multi-class problems

We can imagine the following generalizing SLALOM to multi class problems as follows: We keep an importance map $s : \mathcal{V} \to \mathbb{R}$ that still maps each token to an importance score as previously. However, we now introduce a value score map $v_c : \mathcal{V} \to \mathbb{R}$ for each class $c \in \mathcal{Y}$. Additionally to requiring

$$\sum_{\tau \in \mathcal{V}} s(\tau) = 0. \tag{70}$$

we now require

$$\sum_{c \in \mathcal{Y}} v_c(\tau) = 0, \forall \tau \in \mathcal{V} \tag{71}$$

For an input sequence $\boldsymbol{t}$, the SLALOM model then computes

$$F_c(\boldsymbol{t}) = \log \frac{p(y=1|\boldsymbol{t})}{p(y=0|\boldsymbol{t})} = \sum_{\tau_i \in \boldsymbol{t}} \alpha_i(\boldsymbol{t}) v_c(t_i), \tag{72}$$

The posterior probabilities can be computed by performing a softmax operation over the $F$-scores, as in

$$p(y=c|\boldsymbol{t}) = \frac{\exp(F_c(\boldsymbol{t}))}{\sum_{c' \in \mathcal{Y}} \exp(F_{c'}(\boldsymbol{t}))} \tag{73}$$

We observe that this model has $\big(|\mathcal{Y}| - 1\big)|\mathcal{V}| - 1$ free parameters (for the two-class problem, this yields $2|\mathcal{V}| - 1$ as before) and can be fitted and deployed as the two-class SLALOM without major ramifications.

### C.2  Practical Considerations

Our theoretical model contains slight deviations from real-world transformers to make it amendable to theoretical analysis. To represent token order, common architectures use positional embeddings, tying the embedding vectors to the token position $i$. The behavior that we show in this work's analysis does however also govern transformers with positional embeddings for the following reason: While the positional embeddings could be used by the non-linear ffn part to differentiate sequences of different length in theory, our proofs show that to represent the linear model, the softmax operation must be inverted for any input sequence. This is a highly nonlinear operation and the number of possible sequences grows exponentially at a rate of $|\mathcal{V}|^C$ with the context length $C$. Learning-theoretic considerations (e.g., Bartlett et al., 1998) show that the number of input-output pairs the two-layer networks deployed can maximally represent is bounded by $\mathcal{O}\left(dn_{\text{hidden}} \log(dn_{\text{hidden}})\right)$, which is small ($d{=}786, n_{\text{hidden}}{=}3072$ for BERT) in contrast to the number of sequences ($C = 1024, |\mathcal{V}| \approx 3 \times 10^4$). We conclude that the inversion is therefore impossible for realistic setups and positional embeddings can be neglected, which is confirmed by our empirical findings.

Common models such as BERT also use a special token referred to as CLS-token where the classification head is placed on. In this work, we consider the CLS token just as a standard token in our analysis. In our empirical sections, we always append the CLS token as mandated by the architecture to make the sequences valid model inputs.

## D  Algorithm: Local SLALOM approximation

We propose two algorithms to compute local explanations for a sequence $\boldsymbol{t} = [t_1, \ldots, t_{|\boldsymbol{t}|}]$ with SLALOM scores. In particular, we use the Mean-Squared-Error (MSE) to fit SLALOM on modified sequences consisting of the individual tokens in the original sequence $\boldsymbol{t}$. To speed up the fitting we can sample a large collection of samples offline before the optimization.

### D.1 Efficiently fitting SLALOM with SGD

For the efficient implementation SLALOM-`eff` given in Algorithm 1 we sample minibatches from this collection in each step and perform SGD steps on them. We perform this optimization using $b = 5000$ samples in this work. We use sequences of $n = 2$ random tokens from the sample for SLALOM-`eff`, making the forward passes through the model highly efficient.

### D.2 Fitting SLALOM through iterative optimization

For the high-fidelity implementation SLALOM-`fidel` (Algorithm 2) we first use a different set of sequences and model scores to fit the surrogate: We delete up to 5 tokens from the original sequence randomly to create the estimation dataset (similar to LIME). The fitting algorithm optimized for maximum fidelity uses an iterative optimization scheme to fit SLALOM models. It works by iteratively fitting $\boldsymbol{v}$ and $\boldsymbol{s}$ to the dataset obtained. Denote by $\boldsymbol{f} \in \mathbb{R}^b$ the model scores obtained for the $b$ input sequences $\boldsymbol{t}_i, i = 1...b$. As the SLALOM model in Equation (4) is a linear combination of the values score weighted by the normalized importance score, we can set up a matrix $\boldsymbol{A}$, where element $\boldsymbol{a}_{i,j} = \frac{\exp(s(t_i))}{\sum_{t_j \in t_i} \exp(s(t_j))}$ provides the normalized importance for a given $\boldsymbol{s}$. We solve

$$\min_{\boldsymbol{v}}(\boldsymbol{A}\boldsymbol{v} - \boldsymbol{f})^\top(\boldsymbol{A}\boldsymbol{v} - \boldsymbol{f}), \tag{74}$$

for $\boldsymbol{v}$, which is a linear ordinary least squared problem that can be solved through the normal equation. This results in the optimal $\boldsymbol{v}$ for the given $\boldsymbol{s}$. In a second step, we keep $\boldsymbol{v}$ fixed and find better $\boldsymbol{s}$ scores. We can reformulate the equations for the samples as

$$\sum_{t_j \in \boldsymbol{t}_i} \exp(s(t_j))v(t_j) = \left(\sum_{t_j \in \boldsymbol{t}_i} \exp(s(t_j))\right) \boldsymbol{f}_i \tag{75}$$

$$\Leftrightarrow \sum_{t_j \in \boldsymbol{t}_i} \underbrace{\exp(s(t_j))}_{\bar{s}_j}\underbrace{(v(t_j) - \boldsymbol{f}_i)}_{e_{i,j}} = 0. \tag{76}$$

This problem can be written with a vector $\bar{\boldsymbol{s}} \in \mathbb{R}^{|\mathcal{V}|}$ and a matrix $\boldsymbol{E} \in \mathbb{R}^{b \times |\mathcal{V}|}$ and results in an optimization problem

$$\min_{\bar{\boldsymbol{s}}}(\boldsymbol{E}\bar{\boldsymbol{s}})^\top(\boldsymbol{E}\bar{\boldsymbol{s}}), \tag{77}$$

$$\text{s.t. } \hat{\boldsymbol{s}} \geq \boldsymbol{0} \tag{78}$$

$$\|\hat{\boldsymbol{s}}\|_1 \geq |\mathcal{V}| \tag{79}$$

The conditions ensure that we can obtain the original $\boldsymbol{s}$-scores as $\log \hat{\boldsymbol{s}}$ (element-wise) and that the trivial solution $\hat{\boldsymbol{s}} = \boldsymbol{0}$ is not assumed. We solve this problem using a solver implemented in `scipy.optimize.least_squares`.

## E Experimental Details

In this section, we provide details on the experimental setups. We provide the full source-code in our GitHub repository.

### E.1 Fitting transformers on a synthetic dataset

#### E.1.1 Dataset construction: Linear Dataset

We create a synthetic dataset to ensure a linear relationship between features and log odds. Before sampling the dataset, we fix a vocabulary of tokens, ground truth scores for each token, and their occurrence probability. This means that each of the possible tokens already comes with a ground-truth score $w$ that has been manually assigned. The tokens, their respective scores $w$, and occurrence probabilities are listed in Table 3. Samples of the dataset are sampled in four steps that are exectued repeatedly for each sample:

---

**Algorithm 1** Local efficient SLALOM approximation (SLALOM-`eff`)

---

**Require:** Sequence $\boldsymbol{t}$, trained model $F$ (outputs log odds), random sample length $n$, learning rate $\lambda$, batch size $r$, sample pool size $b$, number of steps $c$

Initialize $v(t_i) = 0$, $s(t_i) = 0$ $\forall$ unique $t_i \in \boldsymbol{t}$

$B \leftarrow b$ samples of random sequences of length $n$ obtained through uniform sampling of unique tokens in $\boldsymbol{t}$.

Precompute $F(B[i]), i = 1, \ldots, b$   # *perform model forward-pass for each sample in pool*

steps $\leftarrow 0$

**while** steps $< c$ **do**

    $B' \leftarrow$ minibatch of $r$ samples uniformly sampled from the sample pool $B$

    loss $\leftarrow \frac{1}{r} \sum_{k=1}^{r} \left( F(B'[k]) - \text{SLALOM}_{v,s}(B'[k]) \right)^2$ # *compute MSE btw. F and SLALOM using precomputed models outputs $F(B')$*

    $v \leftarrow v - \lambda \nabla_v \text{loss}$ # *Back-propagate loss to update SLALOM parameters*

    $s \leftarrow s - \lambda \nabla_s \text{loss}$

    steps $\leftarrow$ steps $+ 1$

**end while**

**return** $v$, $s - \text{mean}(s)$ # *normalize s to zero-mean*

---

**Algorithm 2** Local high-fidelity SLALOM approximation (SLALOM-`fidel`)

---

**Require:** Sequence $\boldsymbol{t}$, trained model $F$ (outputs log odds), max. number of deletions $n$, learning rate $\lambda$, batch size $r$, sample pool size $b$, number of steps $c$

Initialize $\boldsymbol{s} = 0$, $\boldsymbol{s} = 0$ $\forall$ unique $t_i \in \boldsymbol{t}$

$B \leftarrow b$ samples of random sequences of length $n$ obtained through deleting up to $n$ tokens randomly from $\boldsymbol{t}$.

Precompute $F(B[i]), i = 1, \ldots, b$   # *perform model forward-pass for each sample in pool*

steps $\leftarrow 0$

**while** steps $< c$ **do**

    $\boldsymbol{v} = \arg\min_{\boldsymbol{v}'} \sum_{i=1}^{b} \left( F(B[i]) - \text{SLALOM}_{\boldsymbol{v}',\boldsymbol{s}}(B[i]) \right)^2$ # *Fix $\boldsymbol{s}$ and optimize $\boldsymbol{v}$, OLS problem*

    $\boldsymbol{s} = \arg\min_{\boldsymbol{s}'} \sum_{i=1}^{b} \left( F(B[i]) - \text{SLALOM}_{\boldsymbol{v},\boldsymbol{s}'}(B[i]) \right)^2$ # *Fix $\boldsymbol{v}$ and optimize $\boldsymbol{s}$, Quadratic problem*

    steps $\leftarrow$ steps $+ 1$

**end while**

**return** $\boldsymbol{v}$, $\boldsymbol{s} - \text{mean}(\boldsymbol{s})$ # *normalize s to zero-mean*

---

1. A sequence length $|\boldsymbol{t}| \sim \text{Bin}(p = 0.5, n{=}30)$ is binomially distributed with an expected value of 15 tokens and a maximum of 30 tokens

2. We sample $|\boldsymbol{t}|$ tokens independently from the vocabulary according to their occurrence probability (Table 3)

3. Third, having obtained the input sequence, we can evaluate the linear model by summing up the scores of the individual tokens in a sequence:

$$F(\boldsymbol{t}) = F([t_1, t_2, \ldots, t_{|\boldsymbol{t}|}]) = \sum_{i=1}^{|\boldsymbol{t}|} w(t_i). \tag{80}$$

4. Having obtained the log odds ratio for this sample F(t), we sample the labels according to this ratio. We have $p(y = 1)/p(y = 0) = \exp(F(\boldsymbol{t}))$, which can be rearranged to $p(y = 1) = \frac{\exp(F(\boldsymbol{t}))}{1+\exp(F(\boldsymbol{t}))}$. We sample a binary label $y$ for each sample according to this probability.

The tokens appear independently with the probability $p_{\text{occurrence}}$ given in the table.

### E.1.2 Dataset Construction: SLALOM dataset

We resort to a second synthetic dataset to study the recovery property for the SLALOM. This required a SLALOM relation between the data features and the labels. To find a realistic distribution of scores, we

| word | "the" | "we" | "movie" | "watch" | "good" | "best" | "perfect" | "ok" | "bad" | "worst" |
|---|---|---|---|---|---|---|---|---|---|---|
| linear weight $w$ | 0.0 | 0.0 | 0.0 | 0.0 | 0.6 | 1.0 | 1.5 | -0.6 | -1.0 | -1.5 |
| $p_{\text{occurrence}}$ | 1/6 | 1/6 | 1/6 | 1/6 | 1/15 | 1/20 | 1/20 | 1/15 | 1/20 | 1/20 |

Table 3: Tokens in the linear dataset with their corresponding weight

compute a BoW importance scores for input tokens of the BERT model on the IMDB dataset by counting the class-wise occurrence probabilities. We select 200 tokens randomly from this dataset. We use these scores as value scores $v$ but multiply them by a factors of 2 as many words have very small BoW importances. In realistic datasets, we observed that value scores $v$ are correlated with the importance scores $s$. Therefore, we sample

$$s(\tau) \sim 5\left(v(\tau)^{\frac{3}{2}}\right) + \frac{1}{2}\mathcal{N}(0, 1),\tag{81}$$

which results in the value/importance distribution given in Figure 8. We assign each word an equal occurrence probability and sample sequences of words at uniformly distributed lengths in random $[1, 30]$. After a sequence is sampled, labels are subsequently sampled according to the log odds ratio of the SLALOM.

### E.2   Post-hoc fitting of surrogate models

We train the models on this dataset for 5 epochs, where one epoch contains 5000 samples at batch size of 20 using default parameters otherwise.

For the results in Figure 3, we query the models with sequences that contain growing numbers of the work perfect, i.e. ["perfect", "perfect", ...]. We prepend a CLS token for the BERT models.

For the results in Figure 4(a,b), we then sample 10000 new samples from the linear synthetic dataset (having the same distribution as the training samples) and forward them through the trained transformers. The model log odds score together with the feature vectors are used to train the different surrogate models, linear model, GAM, and SLALOM. For the linear model, we fit an OLS on the log odds returned by the model. We use the word counts for each of the 10 tokens as a feature vector. The GAM provides the possibility to assign each token a different weight according to its position in the sequence. To this end, we use a different feature vector of length $30 \cdot 10$. Each feature corresponds to a token and a position and is set to one if the token $i$ is present at this position, and set to 0 otherwise. We then fit a linear model using regularized (LASSO) least squares with a small regularization constant of $\lambda = 0.01$ because the system is numerically unstable in its unregularized form.

### E.3   Recovering SLALOMs from observations

To obtain the results in Figure 4(c,d), we train the transformer models on $20 \cdot 20000$ samples from the second synthetic dataset (SLALOM dataset). When using a smaller vocabulary size, we only sample the sequences out of the first $|\mathcal{V}|$ possible tokens (but keeping the ground truth scores identical).

### E.4   Training Details for real-world data experiments

**Training details.** In these experiments, we use the IMDB (Maas et al., 2011) and Yelp (Asghar, 2016) datasets to train transformer models on. Specifically, the results in Table 1a are obtained by training 2-layer versions of BERT, DistilBERT and GPT-2 with on 5000 samples from the IMDB dataset for 5 epochs, respectively. We did not observe significant variation in terms of number of layers, so we stick to the simpler models for the final experiments. For the experiments in Table 1b we train and use 6-layer versions of the above models for 5 epochs on 5000 samples of the Yelp dataset. We report the accuracies of these models in Table 4 and additional hyperparameters in Table 5.

**SLALOM vs. Naïve Bayes Ground Truth.** To arrive at the Spearman rank-correlations between SLALOM importance scores $s$, value scores $v$ and their combination $(\exp(s) \cdot v)$ with a ground truth, we fit

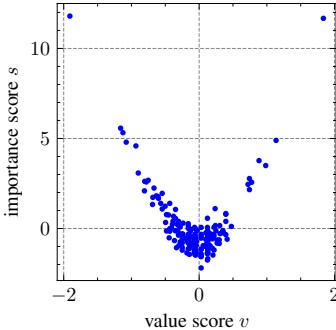

Figure 8: Score distribution for the tokens for the analytical SLALOM used in the recovery experiment.

| dataset | DistilBERT | BERT | GPT-2 |
|---------|-----------|------|-------|
| IMDB | 0.88 | 0.90 | 0.74 |
| Yelp | 0.86 | 0.88 | 0.88 |

Table 4: **Accuracies of models used in this paper.** For IMDB ($|trainset| = 5000$), we use 2-layer versions of the models. For Yelp, ($|trainset| = 5000$), we use 6-layer versions of the models. For both datasets, the $|testset| = 100$. The models are trained for 5 epochs after which we found the accuracy of the model on the test set to have converged.

SLALOM on each of the trained models and use a Naïve-Bayes model for ground truth scores. The model is given as follows:

$$\log \frac{p(y=1|\boldsymbol{t})}{p(y=0|\boldsymbol{t})} = \log \frac{p(y=1)}{p(y=0)} + \sum_{t_i \in \boldsymbol{t}} \log \frac{p(t_i|y=1)}{p(t_i|y=0)} \tag{82}$$

We obtain $\frac{p(t_i|y=1)}{p(t_i|y=0)}$ by counting class-wise word frequencies, such that we obtain a linear score $w$ for each token $\tau$ given by $w(\tau) = \frac{(\#\text{occ. of } \tau \text{ in class 1})+\alpha}{(\#\text{occ. of } \tau \text{ in class 0})+\alpha}$. We use Laplace smoothing with $\alpha = 40$. The final correlations are computed over a set of 50 random samples, where we observe good agreement between the Naïve Bayes scores, and the value and linearized SLALOM scores, respectively. Note that the importance scores are considered unsigned, such that we compute their correlation with the absolute value of the Naive Bayes scores.

**SLALOM vs. Human Attention.** The Yelp Human Attention (HAT) (Sen et al., 2020) dataset consists of samples from the original Yelp review dataset, where for each review real human annotators have been asked to select the words they deem important for the underlying class (i.e. positive or negative). This results in binary attention vectors, where each word either is or is not important according to the annotators. Since each sample is processed by multiple annotators, we use the consensus attention map as defined in Sen et al. (2020), requiring agreement between annotators for a token to be considered important to aggregate them into one attention vector per sample. Since HAT, unlike SLALOM, operates on a word level, we map each word's human attention to each of its tokens (according to the employed model's specific tokenizer).

To compare SLALOM scores with human attention in Table 1b, we choose the AU-ROC metric, where the binary human attention serves as the correct class, and the SLALOM scores as the prediction. We observe how especially the importance scores of SLALOM are reasonably powerful in predicting human attention. Note that the human attention scores are unsigned, such that we also use absolute values for the SLALOM value scores and the linearized version of the SLALOM scores for the HAT prediction.

| parameter | value |
|---|---|
| learning rate | 5e-5 |
| batch size | 5 |
| epochs | 5 |
| dataset size used | 5000 |
| number of heads | 12 |
| number of layers | 2 (IMDB), 6 (Yelp) |
| num. parameter | 31M - 124M |

(a) Hyperparameters

| specification | value |
|---|---|
| CPU core: | AMD EPYC 7763 |
| Num. CPU cores | 64-Core (128 threads) |
| GPU type used | 1xNvidia A100 |
| GPU-RAM | 80GB |
| Compute-Hours | $\approx$ 150 h |

(b) Hardware used (internal cluster)

Table 5: Overview over relevant hyperparameters and hardware

| architecture | $L$ (num.layers) | linear model | GAM | SLALOM |
|---|---|---|---|---|
| GPT-2 | 1 | $20.31 \pm 2.02$ | $48.78 \pm 2.70$ | $\mathbf{16.92} \pm 1.33$ |
| GPT-2 | 2 | $24.81 \pm 3.11$ | $54.33 \pm 3.26$ | $\mathbf{22.17} \pm 1.98$ |
| GPT-2 | 6 | $32.66 \pm 7.60$ | $57.08 \pm 7.19$ | $\mathbf{21.59} \pm 4.14$ |
| GPT-2 | 12 | $25.74 \pm 4.18$ | $54.36 \pm 3.94$ | $\mathbf{20.25} \pm 2.37$ |
| DistilBERT | 1 | $28.28 \pm 4.30$ | $44.43 \pm 2.22$ | $\mathbf{10.83} \pm 2.13$ |
| DistilBERT | 2 | $32.58 \pm 7.75$ | $53.87 \pm 7.20$ | $\mathbf{16.82} \pm 4.38$ |
| DistilBERT | 6 | $31.49 \pm 4.06$ | $49.35 \pm 3.13$ | $\mathbf{17.26} \pm 3.64$ |
| DistilBERT | 12 | $50.82 \pm 9.21$ | $71.64 \pm 9.19$ | $\mathbf{27.50} \pm 4.18$ |
| BERT | 1 | $26.33 \pm 1.90$ | $43.30 \pm 0.88$ | $\mathbf{7.34} \pm 0.70$ |
| BERT | 2 | $28.43 \pm 3.75$ | $48.28 \pm 3.23$ | $\mathbf{9.92} \pm 1.19$ |
| BERT | 6 | $50.82 \pm 6.34$ | $68.23 \pm 4.59$ | $\mathbf{23.99} \pm 3.38$ |
| BERT | 12 | $44.58 \pm 13.15$ | $51.38 \pm 14.71$ | $\mathbf{18.77} \pm 6.78$ |

Table 6: MSE ($\times 100$) when fitting SLALOM to the outputs of transformer models trained on the linear dataset. SLALOM manages to describe the outputs of the transformer significantly better than other surrogate models *even if the underlying relation in the data was linear*.

## F  Additional Experimental Results

### F.1  Fitting SLALOM as a Surrogate to Transformer outputs

We provide an additional emprical counterexample for why GAMs cannot describe the transformer output in Figure 9. The example provides additional intuition for why the GAM is insufficient to describe transformers acting like a *weighted* sum of token importances.

We report Mean-Squared Errors when fitting SLALOM to transformer models trained on the linear dataset in Table 6. These results underline that SLALOM outperforms linear and additive models when fitting them to the transformer outputs. Note that even if the original relation in the data was linear, the transformer does not represent this relation such that the SLALOM describes its output better. We present additional qualitative results for other models in Figure 10 that support the same conclusion.

### F.2  Fitting SLALOM on Transformers trained on data following the SLALOM distribution

We report Mean-Squared Errors in the logit-space and the parameter-space between original SLALOM scores and recovered scores. The logit output are evaluated on a test set of 200 samples that are sampled from the original SLALOM. We provide these quantitative results in Table 7 in Table 8 for the parameter space. In logits the differences are negligibly small, and seem to decrease further with more layers. This finding highlight that a) transformers with more layers still easily fit SLALOMs and such model can be recovered in

| Input Sequence | BERT score | GAM weight assignments | SLALOM weight assignments |
|---|---|---|---|
| perfect | **2.5** | Assign $w_1$("perfect") = 2.5 | Assign $v$("perfect") = 2.5 |
| perfect perfect | **2.5** | Assign $w_2$("perfect") = 0 | Expected SLALOM output: 2.5 |
| worst | **-2.6** | Assign $w_1$("worst") = −2.6 | Assign $v$("worst") = −2.6 |
| worst worst | **-2.5** | Assign $w_2$("worst") = 0.1 | Expected SLALOM output: -2.6 |
| worst perfect | **0.0** | ⚡ Expected GAM output $w_1$("perfect") + $w_2$("worst") = 2.6 | Assign $s$("perfect") = $s$("worst") = 0 Expected SLALOM output: -0.05 |

Figure 9: A simple empirical counterexample for why GANs cannot describe transformer output. We report rounded scores by a real 4-layer BERT model (similar behavior was observed for other layers/architectures) and iteratively fit the GAM $F(\boldsymbol{t}) = \sum_{t_i \in \boldsymbol{t}} w_i(t_i)$ to match observed outputs on the two tokens "perfect" and "worst". We quickly arrive at a contradiction for the GAM. On the contrary, we can assign SLALOM scores that model this behavior with minor error. Because transformers behave like a weighted sum of importances, GAMs are insufficient to model their behavior. In conjunction with Figure 4(a,b) this underlines that GAMs and linear models are insufficient as surrogates.

| $L$ (num.layers) | DistilBERT | BERT | GPT-2 |
|---|---|---|---|
| 1 | $0.002 \pm 0.001$ | $0.002 \pm 0.000$ | $0.011 \pm 0.009$ |
| 2 | $0.003 \pm 0.002$ | $0.003 \pm 0.002$ | $0.017 \pm 0.011$ |
| 6 | $0.001 \pm 0.001$ | $0.011 \pm 0.007$ | $<0.001 \pm 0.000$ |
| 12 | $<0.001 \pm 0.000$ | $<0.001 \pm 0.000$ | $<0.001 \pm 0.000$ |

Table 7: MSE ($\times 100$), logit space, averaged over 5 runs

parameters space. The results on the MSE in parameter space show no clear trend, but are relatively small as well (with the largest value being MSE=0.015 (note that results in the table are multiplied by a factor of 100 for readability). Together with our quantitative results in Figure 4(c,d), this highlights that SLALOM has effective recovery properties.

### F.3 Additional Results on Real-World Data

We obtain SLALOM explanations for real-world data using the procedure outlined in Algorithm 1 (SLALOM-`eff`) with sequences of length $n = 2$ and with Algorithm 2 (SLALOM-`fidel`) removing up to 5 tokens that we compare with Naive-Bayes scores and Human Attention.

### F.3.1 Additional Qualitative Results

Figure 12 shows the full results from the sample used in Figure 5, where we only visualized a choice of words for readability purposes. After running SLALOM-`eff` on our trained IMDB models, we use to explain a movie review taken from the dataset, visualizing value scores $v$ against importance scores $s$.

| $L$ (num.layers) | DistilBERT | BERT | GPT-2 |
|---|---|---|---|
| 1 | $0.092 \pm 0.045$ | $0.540 \pm 0.301$ | $0.940 \pm 0.392$ |
| 2 | $0.094 \pm 0.049$ | $0.368 \pm 0.085$ | $1.652 \pm 0.903$ |
| 6 | $0.124 \pm 0.030$ | $0.830 \pm 0.182$ | $0.569 \pm 0.177$ |
| 12 | $0.287 \pm 0.088$ | $0.394 \pm 0.255$ | $0.385 \pm 0.126$ |

Table 8: MSE ($\times 100$), parameter space, averaged over 5 runs

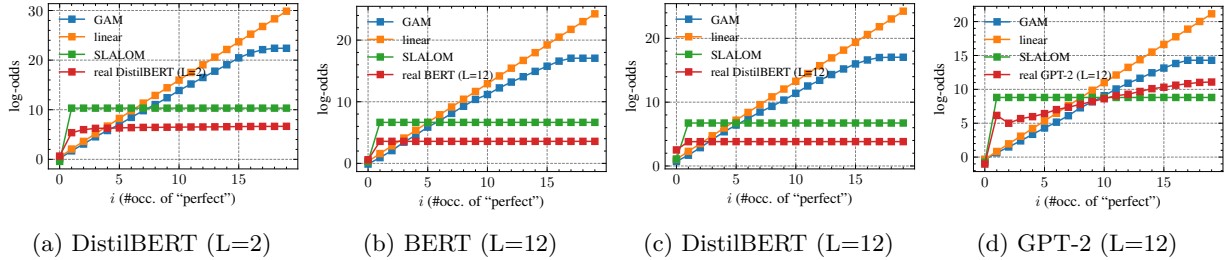

|                          |                          |                          |                          |
| (a) DistilBERT (L=2)     | (b) BERT (L=12)          | (c) DistilBERT (L=12)    | (d) GPT-2 (L=12)         |

Figure 10: **SLALOM describes outputs of transformer models well.** Fitting SLALOM to the outputs of the models shown in Figure 3 using the synthetic dataset. We show results for a sequence containing repetitions of the token "perfect". Note however that the models were trained on a larger dataset of random sequences samples as described in Appendix E.1.1, but these sequences were chosen for visualization purposes. Results on additional models. Despite having $C/2=15\times$ more parameters than the SLALOM model, the GAM model does not describe the output as accurately. We provide quantitative results in Table 6.

|            | SLALOM-`fidel` | | | SLALOM-`eff` | | |
| LM | values $v$ | importances $s$ | lin. SLALOM | values $v$ | importances $s$ | lin. SLALOM |
|---|---|---|---|---|---|---|
| DistilBERT | $0.602 \pm 0.10$ | $0.020 \pm 0.08$ | $0.602 \pm 0.10$ | $0.692 \pm 0.05$ | $0.373 \pm 0.09$ | $0.693 \pm 0.05$ |
| BERT | $0.475 \pm 0.12$ | $0.031 \pm 0.09$ | $0.474 \pm 0.12$ | $0.619 \pm 0.08$ | $0.349 \pm 0.09$ | $0.626 \pm 0.08$ |
| GPT-2 | $0.467 \pm 0.17$ | $0.017 \pm 0.08$ | $0.468 \pm 0.17$ | $0.618 \pm 0.08$ | $0.292 \pm 0.10$ | $0.619 \pm 0.08$ |

| LM | LIME | SHAP | IG | Grad | LRP |
|---|---|---|---|---|---|
| Distilbert | $0.691 \pm 0.05$ | $0.619 \pm 0.06$ | $-0.285 \pm 0.12$ | $-0.215 \pm 0.12$ | $0.706 \pm 0.05$ |
| BERT | $0.616 \pm 0.08$ | $0.554 \pm 0.09$ | $-0.125 \pm 0.14$ | $-0.123 \pm 0.14$ | $0.639 \pm 0.08$ |
| GPT2 | $0.213 \pm 0.13$ | $0.560 \pm 0.09$ | $0.033 \pm 0.13$ | $0.031 \pm 0.13$ | $0.615 \pm 0.08$ |

Table 9: Correlation with linear Naive Bayes Scores. The scores obtained with SLALOM-`eff` (value, lin.) are better than those from LIME, SHAP and comparable to LRP scores.

### F.3.2    Correlation with Naive-Bayes Scores

We compare the scores obtained with SLALOM with the scores obtained with other methods in Table 9, obtaining scores that are reliable with SLALOM-`eff` (value scores and linear scores) in particular. While SHAP achieves higher correlation on BERT, SLALOM achieves higher correlation than LIME and SHAP on all models and higher correlations than LRP for GPT-2 while obtaining slightly inferior values for the BERT-based architectures.

### F.3.3    Human Attention

In Figure 11, we show qualitative results for a sample from the Yelp-HAT dataset. After fitting SLALOM on top of the resulting model, we can extract the importance scores given to each token in the sample. We can see that the SLALOM scores manage to identify many of the tokens which real human annotators also deemed important for this review to be classified as positive. We also show qualitative results for the other methods. However, we suggest caution when interpreting explanations visually without ground truth. We argue that (1) theoretical properties of explanations (2) comparing to a known ground truth as well as (3) consideration of metrics from different domains, e.g., faithfulness, human perspective, are required to allow for a comprehensive evaluation. This is the approach taken in our work.

We show a quantitative comparison of the scores obtained with SLALOM with the scores obtained with other methods on the comparison with Human-Attention in Table 10.

### F.4    Insertion and Removal Benchmarks

It is important to verify that SLALOM scores are competitive to other methods in classical explanation benchmarks as well. We therefore ran the classical removal and insertion benchmarks with SLALOM

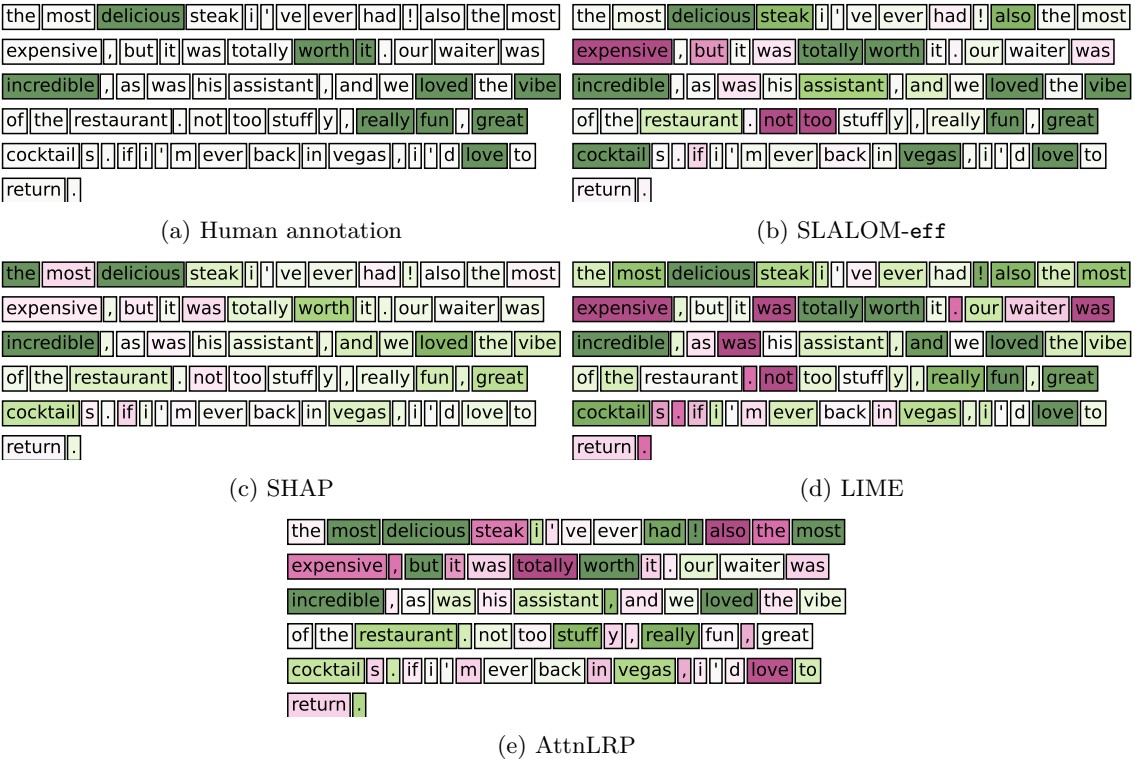

Figure 11: **Qualitative comparisons of attribution maps.** We provide attribution maps for the different techniques in this figure. Many words deemed important by human annotators are likewise highlighted by SLALOM and other techniques.

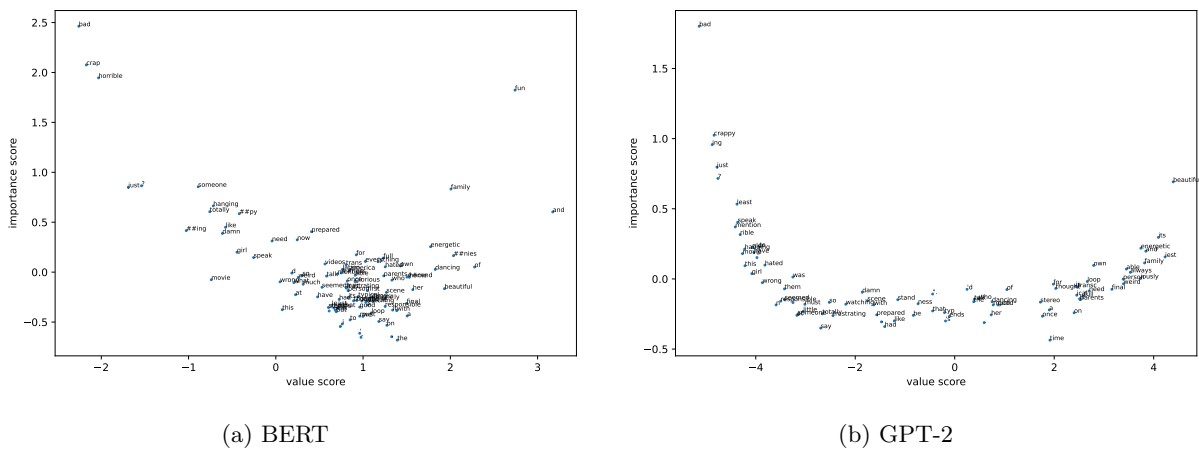

Figure 12: Full scatter plots of SLALOM scores for the sample shown in the main paper (please zoom in for details). We observe that words like "bad" or "fun" get assigned high importance scores and value scores of high magnitude (albeit with different signs) by SLALOM.

| LM | values $v$ | importances $s$ | lin. | LIME | SHAP | LRP |
|---|---|---|---|---|---|---|
| Bert | $0.786 \pm 0.01$ | $0.807 \pm 0.01$ | $0.801 \pm 0.01$ | $0.805 \pm 0.01$ | $0.800 \pm 0.01$ | $0.813 \pm 0.01$ |
| Distil-BERT | $0.688 \pm 0.01$ | $0.681 \pm 0.01$ | $0.686 \pm 0.01$ | $0.702 \pm 0.01$ | $0.668 \pm 0.01$ | $0.703 \pm 0.01$ |
| GPT-2 | $0.674 \pm 0.01$ | $0.685 \pm 0.01$ | $0.683 \pm 0.01$ | $0.632 \pm 0.01$ | $0.671 \pm 0.01$ | $0.699 \pm 0.01$ |

Table 10: Comparison of techniques to predict Human Attention. SLALOM-eff (importances) perform better than SHAP, comparably to LIME and slightly below LRP.

| LM | SLALOM-`fidel` $v$-scores | lin. | SLALOM-`eff` $v$-scores | lin. | LIME | SHAP | IG | Grad | LRP |
|---|---|---|---|---|---|---|---|---|---|
| BERT | $0.893 \pm 0.012$ | $0.901 \pm 0.012$ | $0.881 \pm 0.010$ | $0.885 \pm 0.010$ | $0.875 \pm 0.012$ | $0.881 \pm 0.011$ | $0.084 \pm 0.010$ | $0.069 \pm 0.008$ | $0.852 \pm 0.019$ |
| DistilBERT | $0.841 \pm 0.014$ | $0.854 \pm 0.013$ | $0.888 \pm 0.008$ | $0.886 \pm 0.008$ | $0.838 \pm 0.012$ | $0.864 \pm 0.009$ | $0.143 \pm 0.012$ | $0.131 \pm 0.012$ | $0.865 \pm 0.011$ |
| GPT-2 | $0.837 \pm 0.013$ | $0.844 \pm 0.013$ | $0.782 \pm 0.013$ | $0.784 \pm 0.012$ | $0.479 \pm 0.024$ | $0.859 \pm 0.012$ | $0.289 \pm 0.021$ | $0.269 \pm 0.020$ | $0.833 \pm 0.013$ |
| average | $0.857 \pm 0.013$ | $0.866 \pm 0.013$ | $0.851 \pm 0.010$ | $0.852 \pm 0.010$ | $0.731 \pm 0.016$ | $0.868 \pm 0.011$ | $0.172 \pm 0.014$ | $0.156 \pm 0.013$ | $0.850 \pm 0.014$ |

(a) IMDB: Area-Over Perturbation Curve (deletion, higher is better)

| LM | SLALOM-`fidel` $v$-scores | lin. | SLALOM-`eff` $v$-scores | lin. | LIME | SHAP | IG | Grad | LRP |
|---|---|---|---|---|---|---|---|---|---|
| BERT | $0.015 \pm 0.005$ | $0.011 \pm 0.005$ | $0.011 \pm 0.005$ | $0.011 \pm 0.005$ | $0.017 \pm 0.009$ | $0.012 \pm 0.005$ | $0.224 \pm 0.024$ | $0.214 \pm 0.022$ | $0.010 \pm 0.005$ |
| DistilBERT | $0.018 \pm 0.006$ | $0.019 \pm 0.006$ | $0.032 \pm 0.009$ | $0.032 \pm 0.009$ | $0.014 \pm 0.005$ | $0.020 \pm 0.005$ | $0.250 \pm 0.027$ | $0.249 \pm 0.026$ | $0.017 \pm 0.009$ |
| GPT-2 | $0.033 \pm 0.007$ | $0.032 \pm 0.007$ | $0.045 \pm 0.007$ | $0.045 \pm 0.007$ | $0.129 \pm 0.019$ | $0.021 \pm 0.004$ | $0.251 \pm 0.024$ | $0.244 \pm 0.024$ | $0.039 \pm 0.007$ |
| average | $0.022 \pm 0.006$ | $0.021 \pm 0.006$ | $0.029 \pm 0.007$ | $0.029 \pm 0.007$ | $0.053 \pm 0.011$ | $0.018 \pm 0.005$ | $0.242 \pm 0.025$ | $0.236 \pm 0.024$ | $0.022 \pm 0.007$ |

(b) Yelp: Area-Over Perturbation Curve (insertion, lower is better)

| LM | SLALOM-`fidel` $v$-scores | lin. | SLALOM-`eff` $v$-scores | lin. | LIME | SHAP | IG | Grad | LRP |
|---|---|---|---|---|---|---|---|---|---|
| GPT-2 | $0.747 \pm 0.024$ | $0.753 \pm 0.024$ | $0.726 \pm 0.021$ | $0.727 \pm 0.021$ | $0.444 \pm 0.028$ | $0.849 \pm 0.015$ | $0.292 \pm 0.026$ | $0.290 \pm 0.026$ | $0.740 \pm 0.025$ |
| BERT | $0.657 \pm 0.038$ | $0.667 \pm 0.038$ | $0.865 \pm 0.012$ | $0.863 \pm 0.013$ | $0.797 \pm 0.022$ | $0.859 \pm 0.013$ | $0.249 \pm 0.028$ | $0.281 \pm 0.029$ | $0.855 \pm 0.017$ |
| DistilBERT | $0.645 \pm 0.033$ | $0.642 \pm 0.033$ | $0.813 \pm 0.017$ | $0.813 \pm 0.018$ | $0.746 \pm 0.025$ | $0.854 \pm 0.013$ | $0.201 \pm 0.026$ | $0.243 \pm 0.028$ | $0.768 \pm 0.024$ |
| average | $0.683 \pm 0.032$ | $0.687 \pm 0.032$ | $0.801 \pm 0.017$ | $0.801 \pm 0.017$ | $0.663 \pm 0.025$ | $0.854 \pm 0.014$ | $0.247 \pm 0.027$ | $0.271 \pm 0.028$ | $0.788 \pm 0.022$ |

(c) Yelp: Area-Over Perturbation Curve (deletion, higher is better)

Table 11: Additional results for removal/insertion tests: We show results on the IMDB dataset for removal as well as insertion and removal on the Yelp datset.

compared to baselines such as LIME, SHAP, Grad (Simonyan et al., 2013), and Integrated Gradients (IG, Sundararajan et al., 2017). For the insertion benchmarks, the tokens with the highest attributions are inserted to quickly obtain a high prediction score to the target class. For the deletion benchmark, the tokens with the highest attributions are deleted from the sample to obtain a low score for the target class. We subsequently delete/insert more tokens and compute the "Area Over the Perturbation Curve" (AOPC) as in DeYoung et al. (2020), which should be high for deletion and low for insertion. In addition to the insertion results in Table 1c, the removal results are shown in Table 11a. We show results for the Yelp dataset in Table 11b and Table 11c. We claim that linear SLALOM scores perform on par with LIME and SHAP in this benchmark, but do not always outperform them in this metric. For surrogate techniques (LIME, SHAP, SLALOM) we use 5000 samples each.

## F.5 Error Analysis for non-transformer models

We also investigate the behavior of SLALOM for models that do not precisely follow the architecture described in the Analysis section of this paper. In the present work, we consider an attribution method that is specifically catered towards the transformer architecture, which is the most prevalent in sequence classification. We advise caution when using our model when the type of underlying LM is unknown. In this case, model-agnostic interpretability methods may be preferred.

However, we investigate this issue further: We applied our SLALOM-`eff` approach to a simple, non-transformer sequence classification model on the IMDB dataset, which is a three-layer feed-forward network based on a

|  | SHAP | LIME | lin. SLALOM-`eff` |
|---|---|---|---|
| Insertion (lower) | $0.0120 \pm 0.005$ | $0.0053 \pm 0.003$ | $0.0039 \pm 0.001$ |
| Deletion (higher) | $0.8022 \pm 0.026$ | $0.9481 \pm 0.005$ | $0.9601 \pm 0.004$ |

Table 12: AOPC explantion fidelity metrics for the Fully Connected TF-IDF model. The scores highlight that SLALOM can also provide faithful explanations for non-transformer models due to its general expressivity.

(a) DistilBERT

| Approach / # samples | 1000 | 2000 | 5000 | 10000 |
|---|---|---|---|---|
| SHAP | $2.35 \pm 0.01$ | $4.62 \pm 0.02$ | $11.56 \pm 0.03$ | $23.08 \pm 0.08$ |
| LIME | $0.80 \pm 0.04$ | $1.58 \pm 0.07$ | $3.93 \pm 0.19$ | $8.04 \pm 0.39$ |
| SLALOM-`fidel` | $0.74 \pm 0.03$ | $1.42 \pm 0.06$ | $3.77 \pm 0.24$ | $7.95 \pm 0.41$ |
| SLALOM-`eff` | $0.42 \pm 0.01$ | $0.80 \pm 0.01$ | $2.03 \pm 0.01$ | $4.13 \pm 0.02$ |
| LRP | $0.02 \pm 0.00$ | $0.02 \pm 0.00$ | $0.02 \pm 0.00$ | $0.02 \pm 0.00$ |
| IG | $0.02 \pm 0.00$ | $0.02 \pm 0.00$ | $0.02 \pm 0.00$ | $0.02 \pm 0.00$ |
| Grad | $0.01 \pm 0.00$ | $0.01 \pm 0.00$ | $0.01 \pm 0.00$ | $0.01 \pm 0.00$ |

(b) BLOOM-7B

| Runtime (s) | SHAP | LIME | IG | Grad | SLALOM-`eff` |
|---|---|---|---|---|---|
| 1000 Samples | $25.04 \pm 2.37$ | $15.73 \pm 2.37$ | $0.29 \pm 0.04$ | $0.06 \pm 0.01$ | $0.62 \pm 0.02$ |

Table 13: Runtime results for computing different explanations. SLALOM is substantially more efficient that other surrogate model explanations (e.g., LIME, SHAP). Gradient-based explanations can be computed even quicker, but are very noisy and require backward passes. Runtimes are given in seconds (s).

TF-IDF representation of the inputs. We compute the insertion and deletion Area-over-perturbation-curve metrics that are given in Table 12.

These results show that due to its general expressivity, the SLALOM model also succeeds to provide explanations for non-transformer models that outperform LIME and SHAP in the removal and insertion tests. We also invite the reader to confer Table 1b, where we show that SLALOM can predict human attention for large models, including the non-transformer Mamba model (Gu & Dao, 2023).

## F.6 Runtime analysis

We ran SLALOM as well as other feature attribution methods using surrogate models and compared their runtime to explain a single classification of a 6-layer BERT model. We note that the runtime is mainly determined by the number of forward passes to obtain the samples to fit the surrogates. While this number is independent of the dataset size, longer sequences require more samples for the same approximation quality. The results are shown in Table 13.

While IG and Gradient explanations are the quickest, they also require backward passes which have large memory requirements. As expected, the computational complexity for surrogate model explanation (LIME, SHAP, SLALOM) is dominated by the number of samples and forward passes done. **Our implementation of SLALOM is around 2x faster than LIME and almost 5x faster than SHAP** (all approaches used a GPU-based, batching-enabled implementation), which we attribute to the fact that SLALOM can be fitted using substantially shorter sequences than are used by LIME and SHAP.

We are interested to find out how many samples are required to obtain an explanation of comparable quality to SHAP. We successively increase the number of samples used to fit our surrogates and report the performance in the deletion benchmark (where the prediction should drop quickly when removing the most important tokens). We report the Area over the Perturbation Curve (AOPC) as before (this corresponds to their

| Number of samples | Deletion AOPC |
|---|---|
| SHAP (nsamples="auto") | $0.9135 \pm 0.0105$ |
| SLALOM, 500 samples | $0.9243 \pm 0.0105$ |
| SLALOM, 1000 samples | $0.9236 \pm 0.005$ |
| SLALOM 2000 samples | $0.9348 \pm 0.005$ |
| SLALOM, 5000 samples | $0.9387 \pm 0.005$ |
| SLALOM, 10000 samples | $0.9387 \pm 0.005$ |

Table 14: Ablation study on the number of samples required to obtain good explanations. The results highlight that a number as low as 500 samples can be sufficient to fit the surrogate model at a quality comparable to SHAP.

Comprehensiveness metric of ERASER (DeYoung et al., 2020), higher scores are better). We compare the performance to `shap.KernelExplainer.shap_values(nsamples=auto)` method of the shap package in Table 14. Our results indicate that sampling sizes as low as 500 per explained instance (which is as low as predicted by our theory, with average sequence length of 200) already yields competitive results.

### F.7 Applying SLALOM to Large Language Models

Our work is mainly concerned with sequence classification. In this application, we observe mid-sized models like BERT to be prevalent. On the huggingface hub, among the 10 most downloaded models on huggingface, 9 are BERT-based and the remaining one is another transformer with around 33M parameters[3] (as of September 2023). In common benchmarks like DBPedia classification[4], the top-three models are transformers with two of them also being variants of BERT. We chose our experimental setup to reflect this. Nevertheless, we are interested to see if SLALOM can provide useful insights for larger models as well and therefore experiment with larger models. To this end, we use a model from the BLOOM family (Le Scao et al., 2023) with 7.1B parameters as well as the recent Mamba model (2.8B) (Gu & Dao, 2023) on the Yelp-HAT dataset and compute SLALOM explanations. Note that the Mamba model does not follow the transformer framework considered in this work. We otherwise follow the setup described in Figure 3 and assess whether our explanations can predict human attention. The results on the bottom of Table 1b highlight that this is indeed the case, even for larger models. The ROC scores are in a range comparable to the ones obtained for the smaller models. For the non-transformer Mamba model we observe a drop in the value of the importance scores. This may suggest that value scores and linearized SLALOM scores are more reliable for large, non-transformer models.

**Applying SLALOM to blackbox models.** Finally, we would like to emphasize that SLALOM, as a surrogate model can be applied to black-box models as well. To impressively showcase this, we apply SLALOM to OpenAI's GPT-4 models via the API (Appendix F.7). We use the larger GPT-4-turbo and smaller GPT-4o-mini for comparison. We prompt the model with the following template to classify the review and only output either 0 or 1 as response.

---
**SYSTEM:** You are assessing movie reviews in an online forum. Your goal is to assess the reviews' overall sentiment as 'overall negative' (label '0') or 'overall positive' (label '1'). Your will see a review now and you will output a label. Make sure to only answer with either '0' or '1'.
**USER:** <the review>

---

We then use the token probabilities returned by the API to compute the score. We create 500 samples as a training dataset for the surrogate model and fit SLALOM the model using SLALOM-eff. We obtain the importance plots shown in Figure 13. This highlight that SLALOM scales up to large models. However, we would like to stress that there can be no guarantees as we have no knowledge about the specific structure of the model.

---

[3]`https://huggingface.co/models?pipeline_tag=text-classification&sort=downloads`
[4]`https://paperswithcode.com/sota/text-classification-on-dbpedia`

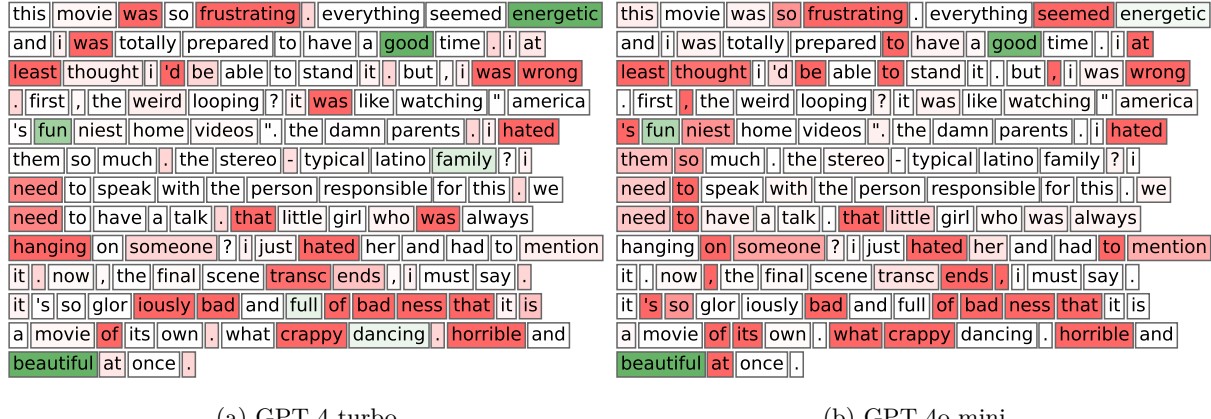

(a) GPT-4-turbo        (b) GPT-4o-mini

Figure 13: We apply SLALOM to OpenAI GPT-4-turbo and GPT-4o to showcase its scalability. We use SLALOM-`eff` scores and linearize them to obtain the highlighted attributions.

### F.8   Case study: Identifying Vulnerabilites and Weaknesses in classification models with SLALOM

In our public code repository, we provide a case study to highlight how SLALOM can be used to uncover practical flaws in transformer classification models. We use a trained BERT model on IMDB sentiment classification as an example and compute SLALOM scores for the tokens in the first 15 samples of the test set ($\approx$ 1200 tokens). SLALOM's scores make it easy to see which tokens have a potentially severe impact on the movie review outcome. We first select all tokens with an importance score $> 2$, all other words have no substantial impact, in particular when added to a larger sequence. We provide a visualization of these tokens in Figure 14. We then make some interesting discoveries mainly based on value scores:

1. There are more words that are negatively interpreted by the model than positive words

2. Out of the words that have highly negative value scores (+importance $>2$), we identify several words that are some that are not directly negatively connotated, e.g., "anyway", "somehow", "never", "anymore", "probably", "doesn", "maybe", "without", "however", "surprised"

We then show that by a few (4) minor modification steps, e.g., adding some of these words to a review, we can change the classification decision for a review from positive to negative, without essentially altering its content (i.e., we manually construct an adversarial example.) This highlights how SLALOM can intuitively help to uncover 1) the influential tokens that contribute most to the decision and 2) allow for a fine-grained analysis that can help uncover spurious concepts and give practitioners an intuitive understanding of a model's weaknesses.

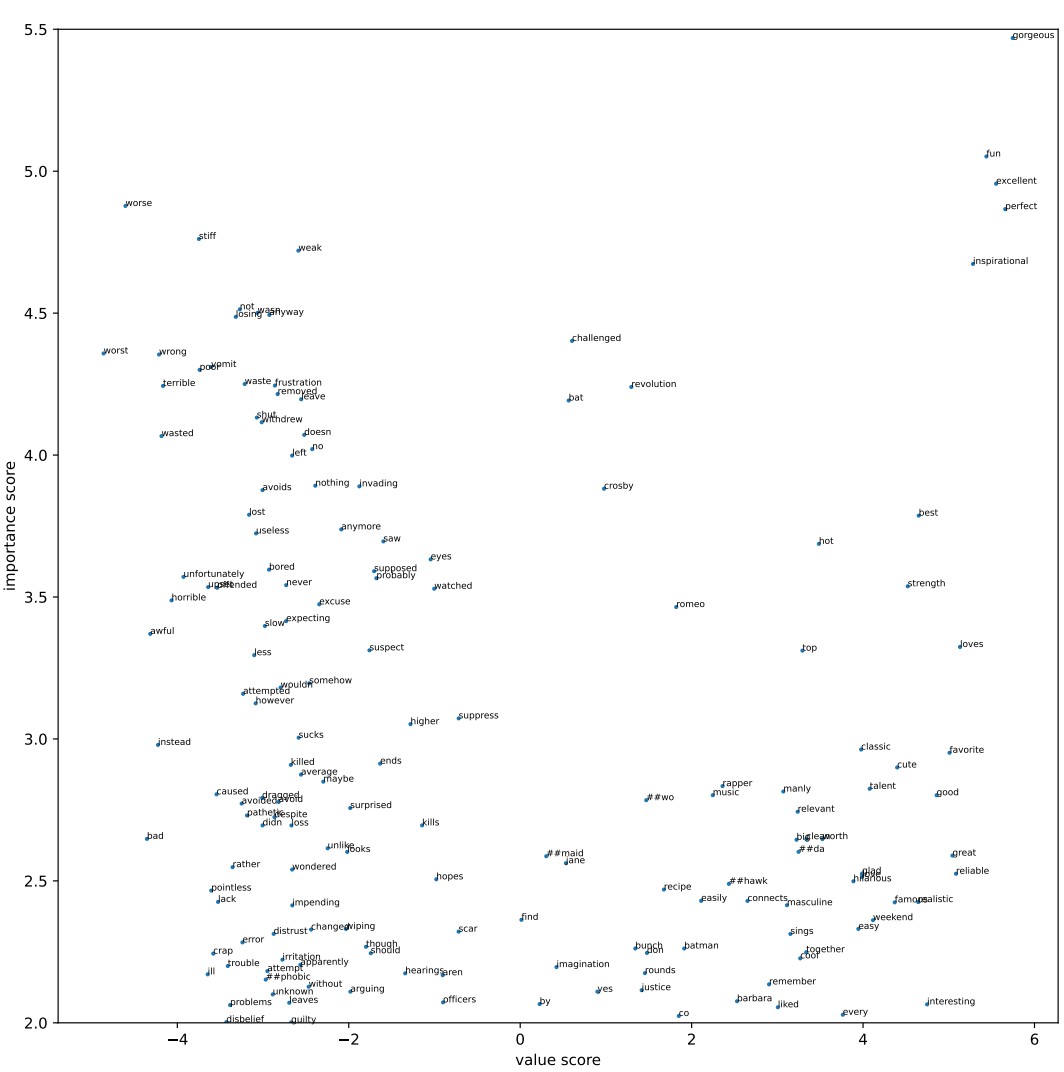

Figure 14: High-impact tokens for BERT identified in our case study. Use zoom for best view.

