# OpenReview forum: "Attention Mechanisms Don’t Learn Additive Models: Rethinking Feature Importance for Transformers"
_TMLR — Accepted by TMLR_

### Review · Reviewer_qMY8 · 2024-11-12

**Summary Of Contributions:**

**Summary:**

This paper addresses the limitations of traditional feature attribution methods in explaining the output of transformer models, which are widely used in natural language processing and other fields. Specifically, it demonstrates the incompatibility between transformer architectures and linear or additive surrogate models, highlighting that transformers are structurally unable to align with these traditional models.

**Contributions:**

The authors introduce the Softmax-Linked Additive Log-Odds Model (SLALOM) as a new surrogate model. SLALOM is designed to work with the transformer framework, aiming to provide more accurate and computationally efficient explanations. The authors claim that SLALOM provides higher fidelity explanations with less computational cost than existing methods and validate this claim through synthetic and real-world datasets.

**Audience:**

Yes

**Claims And Evidence:**

Yes

**Requested Changes:**

For a more holistic interpretability approach, provide examples or scenarios where SLALOM’s two-dimensional token representation enhances practical understanding for non-technical stakeholders. For instance, a case study or visualization showing how SLALOM can help end-users interpret a model's decision would be valuable (not in the main text, in the public repo, or appendix).

The paper mentions that SLALOM performs efficiently on smaller models but lacks detail on scalability in large transformer-based models, which are often deployed in real-world applications. Testing SLALOM on models larger than 7B parameters (if feasible) or discussing potential scaling strategies could be insightful.

**Strengths And Weaknesses:**

**Strengths:**

The introduction of SLALOM offers a good approach to model interpretability within the transformer framework. By addressing a known issue with traditional attribution methods, this model can push forward the applicability of explainability in complex domains where transformers are widely used.

The authors provide rigorous proofs demonstrating the fundamental limitations of using linear or additive models as surrogate explanations for transformers. These proofs add a strong theoretical backing to the study, differentiating it from more empirical studies.

Experiments are performed on synthetic and real-world datasets to show the efficacy of SLALOM, especially its efficiency in high-fidelity explanations compared to other methods like LIME and SHAP.


**Weaknesses:**

SLALOM operates at the token level, assigning individual importance and value scores. However, it lacks mechanisms for contextual or higher-order interpretability, which are crucial in real-world NLP tasks where the meaning and impact of phrases, clauses, or sentences as a whole are often more significant than isolated tokens.


In transformer models, tokens can sometimes gain high importance due to dataset-specific correlations that do not generalize well. SLALOM’s reliance on token-level importance scores might inadvertently amplify these spurious correlations without a mechanism to account for them.

---

> ### Author Response · Authors · 2024-11-30
> **Response to Reviewer qMY8**
>
> We thank the reviewer for the review and for highlighting several positive aspects of our work, including the rigourous proofs that set this work apart from more empirical studies and the experimental evaluation.
>
> We will adress the remaining points below.
>
> > SLALOM operates at the token level, assigning individual importance and value scores. However, it lacks mechanisms for contextual or higher-order interpretability ...
>
> We agree with the reviewer on this inherent limitation of feature attribution methods. There are phenomena that cannot be described on a token or word level. As we point out in our work, we do not believe that there is a one-fits-all explanation and token level methods still have relevant uses (as testified by the large literature on such methods). We have added further discussion regarding this relevant point in our limitations section.
>
> > Tokens can sometimes gain high importance due to dataset-specific correlations that do not generalize well.
>
> Thanks for this point. The proposed method, SLALOM, is designed to accurately reflect the model’s behavior. We therefore hope that SLALOM can be able to help identify exactly such correlations if present in the model. One such case is exemplified in our new case study.
>
> >  [...] Provide examples or scenarios where SLALOM’s two-dimensional token representation enhances practical understanding [...]
> Acase study or visualization showing how SLALOM can help end-users interpret a model's decision would be valuable
>
> We have added a Jupyter notebook with a case-study (```notebooks/CaseStudy.ipynb```) showing how SLALOM can help identifying vulnerabilities and spurious correlations in classification models. We use a trained BERT model on IMDB sentiment classification as an example and compute SLALOM scores for the words in the first 15 samples of the test set (~1200 words). SLALOM’s scores make it easy to see which tokens have potentially severe impact on the movie review outcome. We therefore first select all tokens with an importance score > 2, all other words have no substantial impact, in particular when added to a larger sequence. We then visualize these tokens (we have provided this visualization for completeness in Figure 13 in the Appendix of the revised paper). We make some interesting discoveries mainly based on value scores:
>
> * There are more words that are negatively interpreted by the model than positive words
> * Out of the words that have highly negative value scores (+importance >2), we identify several words that are some that are not directly negatively connotated, e.g., “anyway”, “somehow”, “never”, “anymore”, “probably”, “doesn”, “maybe”, “without”, “however”, “surprised”, …
>
> We then show that by a few (4) minor modification steps identified through the analyis with SLALOM, e.g., adding some of these words to a review, we can change the classification decision from positive to negative, without essentially altering its content (i.e., we manually construct an adversarial example). This highlights how SLALOM can intuitively help to uncover 1) the influential tokens that contribute most to the decision via importance scores and 2) allow for a fine-grained analysis that can help uncover spurious concepts using values scores and give practitioners an intuitive understanding of a model’s weaknesses.
>
> > The paper mentions that SLALOM performs efficiently on smaller models but lacks detail on scalability in large transformer-based models
>
> Our work is mainly concerned with sequence classification. In this application, we observe mid-sized models like BERT to be prevalent. On the huggingface hub, among the 10 most downloaded models on huggingface, 8 are BERT-based [1]. This is reflected in our experimental evaluation.
>
> We would like to stress that SLALOM is a surrogate technique, so it can be applied to any model. The runtime is mainly determined by the time taken in the forward passes to sample a dataset of input/output pairs. Thus, SLALOM scales with the inference time of the predictive model. If inference is possible, running SLALOM is also possible (but requires more passes). To impressively showcase this, we apply SLALOM to OpenAI’s GPT-4 models via the API (Appendix F.7). We use GPT-4-turbo and GPT-4o-mini. We prompt the model with a template (full template in the Appendix) to classify the review and only output either 0 or 1 as response. We then use the token probabilities returned by the API to compute the score. We create 500 samples as a training dataset and fit a SLALOM model using SLALOM-eff. We obtain the importance plots shown in Figure 13 (Appendix) that show reasonable attribution patterns and highlight the favorable scalability characteristics of SLALOM.
>
> ---------------------
>
> We thank the reviewer again and sincerely hope that we have addressed the remaining points. We are happy to answer further questions.
>
> **References**
>
> [1] https://huggingface.co/models?pipeline_tag=text-classification&sort=downloads

---

> > ### Author Response · Authors · 2024-12-13
> > **Let us know if you have any additional questions.**
> >
> > Dear reviewer,
> > thank you again for your constructive feedback. We did our best to incorporate your suggestions into the manuscript. As the discussion period slowly comes to a close, we are eager to know whether your points are addressed, our whether you have any additional requests. In that case, please let us know.

---

> > > ### Comment · Reviewer_qMY8 · 2024-12-16
> > >
> > > I appreciate the authors' reply and the case study notebook. It is very helpful, and my concerns have been addressed.

---

### Review · Reviewer_NUN1 · 2024-11-12

**Summary Of Contributions:**

This paper introduces a novel parameterized surrogate model: **SLALOM** (Softmax-Linked Additive Log-Odds Model), aimed at improving interpretability for transformer architectures. I appreciate the insights offered in this work and find the idea of SLALOM to be a promising approach. I believe the XAI community would be benefited from this research.

The authors argue that traditional feature attribution methods like SHAP and LIME, which rely on the additivity assumption, fail to accurately represent feature importance in transformers due to the complex, context-dependent interactions in their attention mechanisms. However, this motivation fails to convince me. I will offer revision requests later.

To tackle the issue in traditional interpretability methods, such as LIME, SLALOM decomposes the prediction scores into two parts: $F(t) = \sum a_i(t) v(t_i)$ where $a_i(t)$ represents the contextual importance for token $t_i$ and $v(t_i)$ represents the absolute (local) value/importance for token $t_i$, if I understand correctly.


I am convinced by the method, even though the motivation is not properly presented. I think this decomposed interpretability is better to capture token interactions within transformers. The paper provides both theoretical analysis and empirical evidence to support SLALOM’s higher fidelity and computational efficiency, particularly in contrast to SHAP and LIME on synthetic and real-world datasets.

**Audience:**

Yes

**Claims And Evidence:**

Yes

**Requested Changes:**

1.  The paper argues that **additivity** does not yield high fidelity for transformers, where fidelity is crucial for effective explainability. However, in Shapley value theory, **additivity** refers specifically to the marginal contributions of features, not to a simple additive composition of predictions. Further clarification on this distinction is crucial for the soundness of the paper’s argument.
	 - (R1) **I request the authors to re-eaxmine the motivation and their claims in the motivation presentation.**
	- (R2) Also, perhaps the left subfigure of **Figure 1** needed to be revised accordingly.

2. In Section 3.1:
	- (R3) regarding the "**Input and output representations**", the index for the $\mathcal{V}$ should be $i=0,1,\cdots,|\mathcal{V}|$ if I understand correctly. However the manuscript uses $|t|$, which is the number of the input tokens not the length of the vocabulary size.

3. In Section 3.1:
	- (R4) why do we define $E=[e_i]$ as matrix? Will we use it later? If not, I would like to suggest the authors to revise or simply remove it.

4.  In Sections 3.1 and 3.2:
	- (R5) The manuscript seems to confuse the task head for a two-class classification task with the next-token prediction task, which involves predicting across the entire vocabulary. Could you clarify this distinction?

5.  Figure 2:
	- (R6) Figure 2 is somewhat confusing. If the authors intend to illustrate the overall structure of a transformer with multiple layers, it would be helpful to annotate with “\times N” to indicate stacking with N layers, as is common in previous literature. Alternatively, if Figure 2 refers to a single layer only, please clarify this.

6. Section 3.3?
	- (R7) This section seems not to fit in the reading flow. It appears a bit random to me.

7. In Section 4.1:
	- (R8) Could you clarify what is "**The transformer .... identical tokens but different length**". Is a complete sentence? I found this sentence may not be a complete sentence.

8. In section 4.1,
	- (R9) we can not conclude that the weights in Shapley value or attribution analysis are the attention scores. So proposition 4.1 may not hold.

9. In section 4.1,
	- (R10) can the authors clarify the notation $\stackrel{!}{=}$?

10. Section 4.1,
	- (R11) The paragraph in section 4.1 from "The transformer gives rise to ..." is dense and not easy to follow.  I've read several times to understand them. I would like to suggest to polish.

11. Section 4.1,
	- (R12) The conclusion in Section 4.1 doesn't hold since the Positional Encoding. In Transformers, when we add positional embeddings the conclusion doesn't hold. Consider, $p_i$ represents the positional encoding at position $I$, the embedding for token $e_i$, the function $g(e_i, p_i)$ is to encode positional information into the embedding. The $g$ must satisfy some good property (*eg* non-symmetry, *etc*). In this case, the conclusion does not hold. Please clarify.

12. Terms.
	- (R13) Could you please use **map** instead of **mapping** to follow the convention/practice in mathematics?

13. Section 5.
	- (R14) So the work doesn't satisfy the axioms in Shapley value theory. Can you clarify?

14.  Section 5
        - (R15) Can the author formulate how do you the optimize/solve the $v(\tau)$ and $s(\tau)$?

**Strengths And Weaknesses:**

## Strengths
1. The paper is well-written, with a clear and logical flow that makes it easy to follow.
2. The paper has identified the limitations of traditional XAI methods in transformers and present SLALOM to tackle this issue.
3. The evaluation results are convincing, providing the empirical evidence to support SLALOM’s effectiveness.

## Weaknesses
1. The presentation of the major motivation (additivity) fails to convince me. This is because the additivity in Shapley value theory is based on the marginal contributions. For example, in atomic-game in Shapley value theory, regarding $F(t_1, t_2, \cdots, t_k) = \psi_{t_1} + \psi_{t_2} + \cdots + \psi_{t_k}$ where $\psi_{t_i}$ represents the importance of $t_i$, the weights $\psi_{t_i}$ are computed by using the marginal contributions of $t_i$ all coalitions. The motivation, as argued, does not hold theoretically.
2. The paper does not justify the properties in Section 5, which play a crucial role in the paper.
3. Minor concerns: Some figures are not readable and look confusing, for example Figure 3 and Figure 4. I'll give details in the requested revisions.

---

> ### Author Response · Authors · 2024-11-30
> **Response to Reviewer NUN1**
>
> We sincerely thank the reviewer for the detailed feedback and for appreciating the insights provided in our work, leading the reviewer to believe that "the XAI community would be benefited from this research".
>
> We address the reviewer's remaining concerns individually below.
>
> > The presentation of the major motivation (additivity) fails to convince me.
>
> If we understand this point correctly, the reviewer is expressing concerns with our usage of the term “additivity”. We would like to briefly revisit the definitions to make sure there are no misunderstandings in this regard:
> As mentioned by the reviewer, in Shapley value theory, the marginal contributions of each feature sum up to the final score. Definition 1 of the SHAP paper (Lundberg & Lee, 2017) defines additive explanations as:
>
> $$f(x_1, x_2, x_3) = \phi_0 + \sum \phi_i z_i$$
>
> Where $\phi_i$ is the attribution value and $z_i \in {0,1}$ is called a simplified feature (e.g., indicating presence or absence of a value).
> The reviewer is right that we use a different definition in our work: We use the term “additive” as in the literature on Generalized Additive Models (GAMs), defined as (GAM of order 1 in Definition 2 of Bordt et al., 2023).
>
> $$f(x_1, x_2, x_3) = \alpha_i + \sum f_i(x_i)$$
>
> One can prove that Shapley values correspond to the individual contributions, i.e., $f_i(x_i)=\phi_i$, if the model is an order 1 GAM (similar for higher order GAM/Shaply, cf. e.g., Bordt et al., 2023).
>
> We would now like to clarify our motivation: In this work we are interested in explanations that can be used to quantitatively describe model behavior in the neighborhood of an instance, e.g., under input perturbations. Formally, given a model $f$, instance $x$, and explanation $\phi$, we want to describe how $f$ behaves on perturbations of $x$. This is what we refer to as strict fidelity. Therefore, we have to find a way to interpret how the explanation describes local model behavior. The above definition of additivity in Shapley values suggest that a model’s score should decrease by roughly $\phi_i$ when feature $i$ is removed. This is the most natural estimate having Shapley values for $x$ only. We refer to this as a "linear interpretation" of the attribution explanation, which allows us to use the explanation to predict behavior under removals.
>
> We are aware that the value $\phi_i$ is an aggregate over all marginal contributions and that Shapley values can change when the perturbed input is considered, e.g., with a feature removed. We also understand that the Shapley value is not specifically designed for such “perturbation queries”.
>
> This is exactly the gap we would aim to fill with our work: We contribute an explanation that improves strict fidelity for transformers, a property not offered by competing techniques. We have tried to convey this better in our manuscript (Introduction, Related Work Section). We have replaced the wording “additive explanations” in the introduction with “feature attribution” explanation, as the problem is more general.
>
> Still, we are not quite sure if we understood the reviewer’s concern correctly, so we would appreciate further suggestions.
>
> > The paper does not justify the properties in Section 5
>
> Thank you for this comment.  We agree that a discussion at this place is helpful. Our reasoning behind them is as follows:
>
> * Interpretability: This is important as surrogate models are only useful if they are white-box models and their parameters can be directly understood (Molnar, 2021 Chapter 9.2). For instance, boosted trees of higher depth and order are not suitable as surrogate models as they are already too complex to be easily understood by humans.
>
> * Learnability and Recovery. We motivate these two properties in the introduction. Learnability is crucial because using a surrogate model that is hard to represent for the predictive model will likely result in low-fidelity explanations. We show this in our experimental section. Recovery is crucial to re-identify the predictive model’s parameters. Together with learnability, recovery ensures that we can recover relations present in the data which is important, e.g., in scientific discovery with XAI.
> * Efficiency. We believe this is clear, the surrogate model should be efficient to estimate even for larger models. Some other methods, e.g., exact Shapley can be quite inefficient (although we are aware that some reasonably efficient approximations exist).
>
> We hope this clarifies the point. We have added this discussion along with the properties in our revised manuscript (Section 5). Let us know if you have further questions regarding this point.
>
> > Some figures are not readable and look confusing, for example Figure 3 and Figure 4. I'll give details in the requested revisions.
>
> We are happy for suggestions on how to improve readability of the figures, but did not find any remarks regarding Fig. 3 and 4 in the list of requested changes. Please let us know your concerns.

---

> > ### Comment · Reviewer_NUN1 · 2024-12-10
> > **Thanks the efforts of the authors.**
> >
> > Sorry for late replies. I appreciate the efforts from the authors.
> > I truly enjoy the engaged discussions in TMLR venue. I would be beneficial and healthy.

---

> ### Author Response · Authors · 2024-11-30
> **Response to Reviewer NUN1 (cont.)**
>
> We will now detail how we incorporated the requested revisions.
>
> > (R1) I request the authors to re-eaxmine the motivation and their claims in the motivation presentation.
>
> See above
>
> > (R2) Also, perhaps the left subfigure of Figure 1 needed to be revised accordingly.
>
> We agree that for Shapley values, the attribution might be different for the chunks and the concatenated version. However, Figure 1 does not show Shapley values, but values from a linear Naive Bayes model, where the attributions do not change when the sentence is split or not. We have stressed this again in the caption.
>
> > (R3) regarding the "Input and output representations", the index for the V should be i=0,1,⋯,|V| if I understand correctly. However the manuscript uses |t|...
>
> At this passage, we would like to indicate that all individual tokens in the sequence $\mathbf{t}$ should be elements of V. We therefore would like to have $t_i \in \mathcal{V}$ for all $t_i$ in the sequence with indices $i = 1… |t|$. We have rephrased the sentence, let us know if it is clearer now.
>
> > (R4) why do we define  $E=[ei]$  as matrix? Will we use it later?
>
> Thank you for this catch. $E$ is used in the appendix only, so we have moved its definition there accordingly.
>
> > (R5) The manuscript seems to confuse the task head for a two-class classification task with the next-token prediction task, which involves predicting across the entire vocabulary.
>
> In this work, we are interested in the classification task, so the number of outputs should be equal to the number of classes $|\mathcal{Y}|$ as stated the last sentence of Section 3.1. We have mentioned that we are considering transformers for classification in the beginning in Section 3.1. We hope this clarifies the point, let us know if any specific passages require further clarification.
>
> > (R6) Figure 2 is somewhat confusing. If the authors intend to illustrate the overall structure of a transformer with multiple layers, it would be helpful to annotate with $times N$ ...
>
> Figure 2 shows the overall structure with $L$ layers. The Figure includes a $\times L$ in the right corner, where $L$ is defined to be the number of layers in the text. We have highlighted the $\times L$ mark and placed it outside the box for better readability.
>
> > Section 3.3? (R7) This section seems not to fit in the reading flow. It appears a bit random to me.
>
> The purpose of this section is to link our formulation to the practical architectures. We briefly provide the specific parameters used in practical implementation, e.g., the token index r used for the classification head. We have added a clarifying remark regarding this purpose at the start of this section.
>
> > Section 4.1: (R8) Could you clarify what is "The transformer .... identical tokens but different length".
>
> Thank you, we have rewritten this sentence.
>
> > (R9) we can not conclude that the weights in Shapley value or attribution analysis are the attention scores. So proposition 4.1 may not hold.
>
> We are not sure we understand this point correctly, there may be a misconception. The example only shows why transformers struggle to represent additive models on tokens in general. This is independent of the Shapley value. The Shapley value is just one attribution method that can be interpreted in an linear way to derive model behavior under perturbations. Please let us know if we misunderstood this point.
>
> > (R10) can the authors clarify the notation =!
>
> The ! was intended to indicate a requirement, so we require both terms in the equation to be equal. We have however removed this notation now and made this point clear in the text. Thank you for this suggestion.
>
> > (R11) The paragraph in section 4.1 from "The transformer gives [..] is dense and not easy to follow.
>
> Thank you for these points. We have made a thorough pass through this paragraph and hope that it is easier to read now. Please let us know if you have further suggestions.
>
> > (R12) The conclusion in Section 4.1 doesn't hold since the Positional Encoding.
>
> Making transformer models amenable to theoretical analysis requires simplification: The Transformer Circuits thead (https://transformer-circuits.pub/2021/framework/index.html) neglects the MLP part while other works, e.g, von Oswald et al, 2024, neglects the softmax part (which we focus on in this work). We have openly discussed that the model on our analysis is simplified and does not consider positional embeddings (cf. paragraph “Practical Considerations” at the end of section 4.2.). We have added a remark to make this clear earlier into the formalization. We think this is not problematic as long as the results remain valid for real models. Learning-theoretic considerations indicate that even with positional embeddings, commonly-sized models cannot invert the softmax to produce linear output with near certainty. This is impressively confirmed by our findings in Figure 3, obtained on models with positional embeddings. We have added a remark earlier in Section 3.

---

> > ### Author Response · Authors · 2024-11-30
> > **Response to Reviewer NUN1 (cont.)**
> >
> > > (R13) Could you please use map instead of mapping to follow the convention/practice in mathematics?
> > Thanks, we have replaced all occurrences of the term in our manuscript.
> >
> > > (R14) So the work doesn't satisfy the axioms in Shapley value theory. Can you clarify?
> >
> > As our representation is two-dimensional, we don’t think the axioms can be directly applied to SLALOM. However, SLALOM is just another multi-input function, so Shapley values can be computed for it. We have formulated how to compute Shapley values for SLALOM form in Appendix B.7 (basically plugging in SLALOMs form in the Shapley value definition). We have also added a remark in this regard at the end of Section 5.
> >
> >
> > > (R15) Can the author formulate how do you the optimize/solve the v(τ) and s(τ)?
> >
> > Thanks for raising this point. We describe our numerical algorithms in Section 5.3. We have emphasized this again and made the relation to the previous section clearer. Our main algorithm (SLALOM-eff) consists of two steps:
> > * We first sample a dataset of input sequences and corresponding scores by the predictive model. The input sequences are modifications of the instance that we would like to explain. A very simple technique that worked quite well consists of just sampling short sequences of random tokens from the instance.
> > * Our default technique is to find the v and s scores by performing SGD on the dataset created in step (1). We use the mean-squared error between the output from SLALOM and the predictive model to optimize the parameters of SLALOM. We have added a clarifying remark and provide pseudocode and additional description in Appendix D. Please let us know if you would like to see any further clarification.
> >
> > -------------------
> >
> > We thank the reviewer again for the detailed comments. While we did our best to address them, it might be that we have not understood some of the concerns correctly. In this case, we kindly ask the reviewer to clarify their questions. We will be happy to make further changes to fully adress the reviewer’s points.
> >
> > **References**
> >
> > Sebastian Bordt and Ulrike von Luxburg. From shapley values to generalized additive models and back. In
> > International Conference on Artificial Intelligence and Statistics, pp. 709–745. PMLR, 2023.
> >
> > Christoph Molnar. Interpretable Machine Learning. 2019.
> >
> > Von Oswald, Johannes, et al. "Transformers learn in-context by gradient descent." International Conference on Machine Learning. PMLR, 2023.

---

### Review · Reviewer_WXJv · 2024-11-23

**Summary Of Contributions:**

This paper introduces SLALOM, a novel surrogate explanation model designed to address the limitations of Transformers in representing linear and generalized additive models. The authors theoretically and empirically demonstrate the shortcomings of traditional linear attribution methods, such as LIME and SHAP, in aligning with the architecture of Transformers. SLALOM assigns attribution scores by modeling token importance and relevance for each token in an input, converting these into linear attributions comparable to existing methods. Through the Softmax-Linked Additive Log-Odds framework, SLALOM faithfully captures Transformer behaviors and provides explanations aligned with human token annotations. Extensive experiments on synthetic and sentiment classification datasets show that SLALOM outperforms traditional methods in both interpretability and alignment with human annotations.

**Audience:**

Yes

**Broader Impact Concerns:**

no.

**Claims And Evidence:**

Yes

**Requested Changes:**

- Please provide the code.

**Strengths And Weaknesses:**

**Strength**

- Well written paper
- Comparisons are made to common other techniques, (SHAP, LIME, IG, LRP)
- Even runtime times are reported

- Theoretically everything seems solid, with great presentation and consistent notation.

**Weaknesses**
- Experimentation is a bit limited, with not a lot of models considered, but given the theoretical demonstrations, I dont think this is too important, if the code is provided.

**Questions**
- Since you apply SLALOM also to larger models (BLOOM, Mamba), why are these results not included in the main text?
- How can SLALOM theoretically be fit to SSM models, since they are not transformers?

---

> ### Author Response · Authors · 2024-11-30
> **Response to Reviewer WXJv**
>
> We thank the reviewer for their positive review and for seeing the strengths of our work, including the comparisons to relevant techniques and theoretical soundness. We will address the remaining questions below.
>
> > [...] not a lot of models considered, but given the theoretical demonstrations, I dont think this is too important, if the code is provided
>
> While only 3 models are considered in our main paper, we consider additional ones (BLOOM, Mamba and a feed-forward model on TF-IDF in Appendix F.5). We hope that the total of six highly different models is sufficient to provide a realistic picture of the performance of SLALOM across architectures. We added qualitative results with GPT-4 in response to reviewer ``` qMY8```, showing that SLALOM scales up even to the latest models. Please confer our statement regarding code below.
>
> > How can SLALOM theoretically be fit to SSM models, since they are not transformers?
>
> Our paper theoretically considers the classical transformer architecture that is formalized in the text. Thus, we have no formal proof about SLALOM’s alignment with state space models. However, as SLALOM has 2x more parameters than linear models that could be alternatively used, we expect it to be more versatile in general and to also approximate other classes of functions well, including those given by non-transformer models such as Mamba. Our empirical findings support this hypothesis.
>
> > Since you apply SLALOM also to larger models (BLOOM, Mamba), why are these results not included in the main text?
>
> While we think it is an interesting experiment to apply SLALOM to larger models to show that SLALOM does scale up to these models, the main focus of our work lies on text classification tasks, where mid-sized models, mostly BERT derivatives, are prevalent (out of the 10 most downloaded models on Huggingface Hub, 8 are BERT-based [1]). As stated in our previous response, our theoretical results align with classical transformer architecture (e.g., BERT and GPT-2). Our experimental section was designed to reflect these theoretic claims. We choose to not include BLOOM and Mamba in the main text, as we would not like to suggest that our theoretical results cover this case. This study is intended as a proof-of-concept only. As we point out in our limitations section, we recommend caution when applying SLALOM to other architectures, even if technically possible.
>
> However, if the reviewer insists that the results should be moved to the main text, we are happy to do so.
>
> > Please provide the code.
>
> We commit to fully open-sourcing our code. A preliminary version of the code is already publicly visible along with this submission (supplementary material ZIP). We will upload the code along with its corresponding documentation to a public repository (e.g., GitHub), to make it even more accessible and visible. We have added a corresponding remark at the end of the introduction section, where we will add a non-anonymous link in case of acceptance.
>
> ------------------
>
> We thank the reviewer again and are happy to answer any further questions.
>
>
> **References**
>
> [1] https://huggingface.co/models?pipeline_tag=text-classification&sort=downloads

---

> ### Comment · Reviewer_WXJv · 2024-12-01
> **Answer to revisions**
>
> Dear Authors,
>
> Thank you for your detailed responses.
> I have also reviewed the comments and concerns raised by the other reviewers and, in my opinion, the authors have addressed them satisfactorily.
>
> As outlined in my initial review, I have no remaining concerns.
>
> I kindly request that, for publication, you make the code available in a public repository and include a direct link in the abstract.

---

> > ### Author Response · Authors · 2024-12-16
> > **Response to Reviewer WXJv**
> >
> > We are happy to hear that your points have been addressed. In case of acceptance, we commit to linking our code repository in the abstract as suggested.
> >
> > Thank you again for your positive assessment.

---

### Decision · Action_Editor_e8Ro · 2024-12-20

**Recommendation:** Accept with minor revision

**Comment:**

This paper presents SLALOM, a surrogate explanation model designed to explain the outputs of transformers. It addresses the limitations of earlier linear or additive surrogate models like LIME and SHAP, which struggle with transformer architectures due to their attention mechanism. SLALOM tackles this by decomposing attribution scores into two components: token importance and relevance scores for each input token. The paper provides theoretical justification and empirical results. Experiments on synthetic and sentiment classification datasets demonstrate that SLALOM outperforms previous methods.

All reviewers commend the paper's clear writing and strong empirical performance. Reviewer WXJv and qMY8 also emphasize the extensive theoretical analysis provided. However, some concerns were raised. Reviewer WXJv suggests that larger models results be included in the main paper rather than appendix. Reviewer qMY8 raises a concern that relying on token-level importance might limit the method's real-world applicability. Reviewer NUN1 points out issues with the motivation and presentation clarity in Section 5, and Figures 3 and 4.

During the rebuttal process, the authors clarified some misleading points in their writing and provided additional experimental results demonstrating the algorithm's broader applicability. After reviewing the paper, comments, and rebuttals, the action editor believes that most concerns have been addressed in rebuttal and recommends accepting the paper. The authors are advised to revise the paper to improve the writing clarity and include discussions from the rebuttal in the final version.

**Audience:**

Yes.

**Claims And Evidence:**

Yes.

---

> ### Author Response · Authors · 2024-12-30
> **Camera-Ready Version**
>
> Thank you for your positive assessment and your feedback.
> We have uploaded a camera-ready version that incorporates the points mentioned with the changes listed in the summary above.
> Please let us know if you have additional requests.